# ENHANCING LARGE LANGUAGE MODELS IN CODING THROUGH MULTI-PERSPECTIVE SELF-CONSISTENCY

## ABSTRACT

Large language models (LLMs) have exhibited remarkable ability in textual generation. However, in complex reasoning tasks such as code generation, generating the correct answer in a single attempt remains a formidable challenge for LLMs. Previous research has explored solutions by aggregating multiple outputs, leveraging the consistency among them. However, none of them have comprehensively captured this consistency from different perspectives. In this paper, we propose the Multi-Perspective Self-Consistency (MPSC) framework, a novel decoding strategy for LLM that incorporates both inter-consistency across outputs from multiple perspectives and intra-consistency within a single perspective. Specifically, we ask LLMs to sample multiple diverse outputs from various perspectives for a given query and then construct a multipartite graph based on them. With two predefined measures of consistency, we embed both inter- and intra-consistency information into the graph. The optimal choice is then determined based on consistency analysis in the graph. We conduct comprehensive evaluation on the code generation task by introducing solution, specification and test case as three perspectives. We leverage a code interpreter to quantitatively measure the inter-consistency and propose several intra-consistency measure functions. Our MPSC framework significantly boosts the performance on various popular benchmarks, including HumanEval (**+17.60%**), HumanEval Plus (**+17.61%**), MBPP (**+6.50%**) and CodeContests (**+11.82%**) in Pass@1, when compared to original outputs generated from ChatGPT, and even surpassing GPT-4.

## 1 INTRODUCTION

In recent years, pre-trained large language models (LLMs) have demonstrated unprecedented proficiency in understanding, generating, and interacting with human language (Brown et al., 2020; Chowdhery et al., 2022; OpenAI, 2023; Touvron et al., 2023). These models attain remarkable few-shot or even zero-shot performance on a diverse array of natural language processing tasks, ranging from low-level text generation to high-level reasoning and planning, by utilizing vast amounts of textual data during pre-training.

Despite their remarkable abilities, LLMs often struggle to generate the correct answer in a single attempt, especially in complex reasoning tasks like solving mathematical problems or generating code. To address this issue, prior research has sought to aggregate multiple outputs and find answers through their consistency (Wang et al., 2022; Sun et al., 2022; Zhou et al., 2022; Jung et al., 2022). However, these approaches only consider the *intra-consistency* within a single perspective when generating the outputs, potentially leading to a situation akin to the tale of blind men describing an elephant, where LLMs fail to grasp the complete picture and engage in a deliberate thought process.

To avoid this, we further leverage *inter-consistency*, which captures the agreement between generated outputs from diverse perspectives. In this paper, we propose the Multi-Perspective Self-Consistency (MPSC) framework, a novel decoding strategy for LLM that incorporates both inter-consistency across multiple perspectives and intra-consistency within a single perspective. Figure 1a provides a high-level comparison between previous works and our proposed framework. Specifically, we prompt the LLM to generate diverse outputs from multiple perspectives simultaneously, treating them as vertices in a graph. We then establish connections (i.e. edges) based on the pairwise agreement of vertices from different perspectives. Our goal is to identify the most reliable output

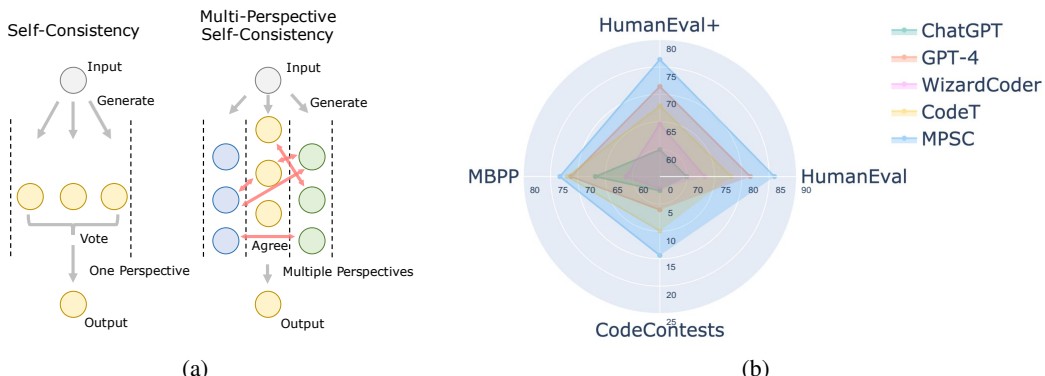

Figure 1: (a) Difference between Self-Consistency and Multi-Perspective Self-Consistency. Self-Consistencsy aggregates diverse outputs from one perspective. We embed both inter- and intra-consistency among multiple perspectives into a graph and leverage them for better inference. (b) The performance of MPSC. With ChatGPT as the foundation model, MPSC even surpasses GPT-4 and achieves SOTA performance on all four benchmarks.

using a score function, which evaluates all vertices by considering both intra- and inter-consistency information encoded in the graph. Specifically, the intra-consistency information guides the function to favor the most internally consistent output within a single perspective, while inter-consistency ensures that the scores for two statements from different perspectives are similar if they reach a consensus. We formalize the learning process of the score function as an optimization problem adhering to these two consistency criteria and leverage an iterative algorithm proposed by Zhou et al. (2003b) to achieve this goal.

To empirically demonstrate the effectiveness of our approach, we apply MPSC in the code generation task. We propose to employ solution, specification and test case as three perspectives to describe a given user intent in natural language. Specifically, solution implements the desired functionality, specification demonstrates the intended properties, while test case outlines the expected behavior for some specific inputs. The intriguing part about these definitions of perspectives is that we can measure the inter-consistency among them deterministically with a code interpreter. We evaluate MPSC on four widely used code generation benchmarks, including HumanEval (Chen et al., 2021), HumanEval+ (Liu et al., 2023), MBPP (Austin et al., 2021) and CodeContests (Li et al., 2022). We employ MPSC to select the most likely correct solutions generated by ChatGPT. Experimental results show that our method boosts the performance by a large margin, 17.60% in HumanEval, 17.61% in HumanEval+, 6.50% in MBPP and 11.82% in CodeContests. Our results even surpass GPT-4 (OpenAI, 2023) as shown in Figure 1b. It's worth noting that our proposed framework is not constrained to code generation. By defining perspectives and designing the corresponding inter- and intra-consistency measures, MPSC can be used for any textual generation scenarios.

## 2 MPSC: MULTI-PERSPECTIVE SELF-CONSISTENCY

A single perspective can often lead to an incomplete understanding of a problem, akin to the parable of "blind men describing an elephant.". The reasoning process of LLMs follows a similar pattern. LLMs generally cannot guarantee the correctness of generated output, especially in complex reasoning tasks requiring comprehensive thinking, even though they possesses the knowledge to produce the correct answer. However, a key aspect of human intelligence is the ability to think from multiple perspectives, resulting in a more comprehensive understanding of situations and more accurate solutions to problems. Inspired by human cognition, we propose a novel decoding strategy for LLMs that enhances their reasoning abilities by incorporating consistency among outputs from multiple perspectives.

## 2.1 GRAPH CONSTRUCTION

We first require the LLM to generate diverse outputs from different perspectives. Perspectives can be defined as different ways to describe a given query. For example, in the context of code generation, we can consider code solutions and test cases as two perspectives, with the former describing the desired functionality and the latter outlining the expected behavior of specific inputs.

We employ graph representations to capture the relationships between diverse outputs. Specifically, we represent them as a set of vertices $V = \{v_1, ..., v_N\}$, where each vertex corresponds to an output. Based on predefined perspectives, we construct an undirected multipartite graph $\mathcal{G} = (V, E)$, with each edge encoding the agreement between a pair of outputs from two distinct perspectives. Our goal is to leverage graphs to encode the multi-perspective consistency information, and then learn a score function $f : V \rightarrow \mathbb{R}$ (also a vector $\boldsymbol{f}$, $f_i = f(v_i)$) from graphs to choose the most reliable answer among all generated outputs.

## 2.2 INCORPORATING MULTI-PERSPECTIVE CONSISTENCY

We distinguish between two kinds of consistency based on the perspectives involved. Intra-consistency is defined as the degree to which a given output aligns with others within the same perspective, following the original definition in Wang et al. (2022). Conversely, inter-consistency is defined as the degree of consensus between a pair of outputs from two different perspectives.

**Inter-Consistency** We draw an analogy with the process of humans acquiring accurate understanding through multiple perspectives. During the investigation of a group crime, police usually interrogate different suspects separately. By comparing testimonies from various perspectives, a more accurate understanding of the case can be achieved. An underlying assumption is that *a pair of statements exhibiting consistency are either both correct or both incorrect*.

To apply the assumption to the constructed graph, we formalize the consistency information by a mathematical representation within the graph by introducing a measure function $\omega : V \times V \rightarrow \mathbb{R}$, which also constructs the adjacency matrix $\boldsymbol{W}$, $W_{i,j} = \omega(v_i, v_j)$. Therefore, the pairwise consistency is represented as edge weights, while the correctness of statements corresponds to the scores of vertices given by $f$. Consequently, the assumption above necessitates minimizing the local variation of $f$ over each edge, expressed as the difference between the scores of two vertices $|f(v_i) - f(v_j)|$.

**Intra-Consistency** Wang et al. (2022) discovered a strong correlation between the intra-consistency of an output and its accuracy. This finding suggests that intra-consistency serves as an estimate of the model's confidence in its generated outputs and reflects correctness to a certain degree. We employ another measure function $\varphi : V \rightarrow \mathbb{R}$ (also a vector $\boldsymbol{y}$, $y_i = \varphi(v_i)$) to quantify the intra-consistency of each vertex in the constructed graph. Subsequently, we can utilize the intra-consistency information as a supervision signal by ensuring the closeness between the score function $f$ and the intra-consistency measure $\varphi$.

## 2.3 OPTIMIZATION FORMULATION

Following the criteria of inter- and intra-consistency, we derive two objectives for each,

$$\mathcal{L}_{inter} = \sum_{(v_i,v_j)\in E} W_{i,j}(f(v_i) - f(v_j))^2 = \boldsymbol{f}^T \boldsymbol{L} \boldsymbol{f} \tag{1}$$

$$\mathcal{L}_{intra} = \frac{1}{2}\sum_{v_i\in V} |f(v_i) - \varphi(v_i)|^2 = \frac{1}{2}\sum_{v_i\in V} |f_i - y_i|^2 = \frac{1}{2}||\boldsymbol{f} - \boldsymbol{y}||^2 \tag{2}$$

where $\boldsymbol{L} = \boldsymbol{D} - \boldsymbol{W}$ is the laplacian matrix of the graph $\mathcal{G}^1$. The objective of inter-consistency is the weighted sum of local variation while the objective of intra-consistency is a Mean Squared Error (MSE). We can then formulate the learning process of $f$ as an optimization problem that combines both $\mathcal{L}_{inter}$ and $\mathcal{L}_{intra}$:

$$\min_{f:V\rightarrow\mathbb{R}} \{\alpha \cdot \mathcal{L}_{inter} + (1 - \alpha) \cdot \mathcal{L}_{intra}\} \tag{3}$$

---

[1]In our experiment, we use the symmetric normalized Laplacian $\boldsymbol{L}^{sym} = \boldsymbol{D}^{-\frac{1}{2}} \boldsymbol{L} \boldsymbol{D}^{-\frac{1}{2}}$ for more robust performance.

To solve this optimization problem on the graph, we adopt the iterative algorithm proposed by Zhou et al. (2003b). The details of the algorithm can be found in Appendix A.

MPSC is highly flexible and extensible. With the specific meaning of perspectives defined, one can plug and play different measure functions of inter- and intra-consistency according to the particular application scenario. In this paper, we take code generation as the target scenario.

# 3 MPSC ON CODE GENERATION

In this section, we illustrate the application of the MPSC framework to a specific scenario, with a focus on the task of code generation. Code generation necessitates proficient natural language understanding, deliberate reasoning and stringent controllability to generate valid solutions. As a result, it has been acknowledged as a fundamental benchmark for evaluating the capabilities of LLMs. Previous works (Li et al., 2022; Chen et al., 2022) utilize test cases as auxiliary verification to improve the quality of generated solutions. Distinct from these approaches, we further propose to incorporate specifications as an additional perspective, inspired by *Formal Verification*. Consequently, we derive a measure of inter-consistency among the three perspectives: *Solution*, *Specification* and *Text case*, which can be deterministically assessed by the python interpreter. Furthermore, we introduce several measures of intra-consistency in terms of varying degrees of agreement, ranging from literal similarity to structural equivalence.

## 3.1 SOLUTION, SPECIFICATION AND TEST CASE

Given a user intent in natural language, we introduce solution, specification and test case as three perspectives to describe the required functionality. A *solution* is the source code implementing the functionality denoted as $g : \mathbb{X} \to \mathbb{Y}$, which is also the target of code generation. A *test case* is a pair of valid input and output satisfying the required functionality denoted as $(x, y) \in \mathbb{X} \times \mathbb{Y}$. *Specification* draws inspiration from *Formal Verification* in software engineering, which mathematically proves the correctness of one program by ensuring its satisfaction of some certain formal specifications. In the context of software engineering, formal verification is usually written in formal programming languages, e.g. Coq (Team, 2023) and Dafny (Leino, 2010), and conducted by accompanying verification tools. For the generalization of the proposed method, we adopt the idea of formal verification and limit the specifications within pre-conditions and post-conditions, which can be written as functions in the same programming language like solutions, without struggling writing formal languages. Specifically, a pre-condition constrains the requirements that a valid input should satisfy, while a post-condition constrains the relationships that a pair of valid inputs and outputs should satisfy. We denote them as $h^{pre} : \mathbb{X} \to \{False, True\}$ and $h^{post} : \mathbb{X} \times \mathbb{Y} \to \{False, True\}$. Detailed examples of outputs from the three perspectives are shown in Figure 2.

## 3.2 INTER-CONSISTENCY MEASURES FOR CODE GENERATION

The most appealing aspect of introducing the three perspectives is that we can use a code interpreter to quantitatively verify the agreement between the outputs from different perspectives, thus obtaining a deterministic measure of inter-consistency.

Consider three vertices, $v_i$ represents an output $g_i \in \{g_1, g_2, ..., g_I\}$ from *Solution* perspective, $v_j$ represents an output $(h_j^{pre}, h_j^{post}) \in \{(h_1^{pre}, h_1^{post}), ..., (h_J^{pre}, h_J^{post})\}$ from *Specification* perspective, and $v_k$ represents an output $(x_k, y_k) \in \{(x_1, y_1), ..., (x_K, y_K)\}$ from *Test case* perspective. We propose to measure the inter-consistency among them as following,

- For *Solution* and *Specification*, $\omega(v_i, v_j) = \mathbb{E}_{x \in \mathbb{X}}[\mathbf{1}_{h_j^{pre}(x) \to h_j^{post}(x, g_i(x))}]$. [2]

- For *Solution* and *Test case*, $\omega(v_i, v_k) = \mathbf{1}_{g_i(x_k)=y_k}$.

- For *Specification* and *Test case*, $\omega(v_j, v_k) = \mathbf{1}_{h_j^{pre}(x_k) \wedge h_j^{post}(x_k, y_k)}$.

We provide the Python code snippets implementing the inter-consistency measure in Appendix B.

---

[2]It is impossible to enumerate all $x \in \mathbb{X}$, thus we use sampling for approximation.

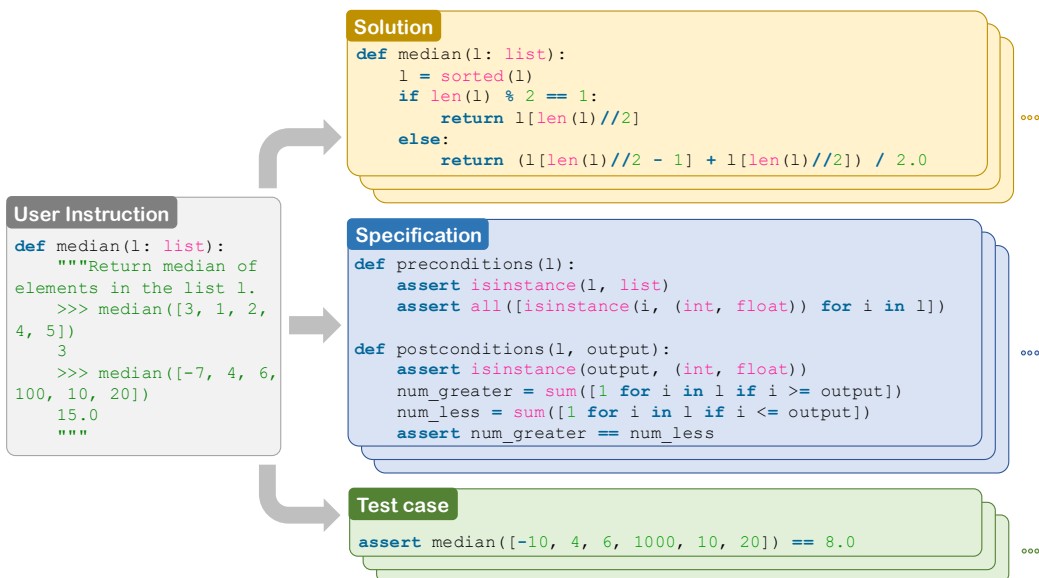

Figure 2: A detailed example of the three perspectives of function `median(l)` from HumanEval.

## 3.3 GENERAL INTRA-CONSISTENCY MEASURES

We also explore several intra-consistency measure functions, as presented in Table 1. These proposed measures are designed to be generalizable across all tasks, as they utilize no prior knowledge and make no assumptions related to the specific perspective definitions.

**Bayes Risk** Minimum Bayes risk decoding (Kumar & Byrne, 2004) selects the output $h \in \mathbb{H}$ that minimizes the expected errors $R(h) = \mathbb{E}_{y \sim P(y)}[L(y, h)]$ over the distribution of label y. Because of the unavailability of $P(y)$, $P(h)$ is usually used as a proxy distribution in practice. Then the Bayes risk can be rewritten as $R(h) = \sum_{h' \in \mathbb{H}} L(h', h) \cdot P(h')$, which is in fact measure the consistency of h over the hypothesis space. Specifically, we use negative BLEU metrics (Papineni et al., 2002) as loss function for and assume that the hypothesis space is uniform for generality.

**Structural Equivalence** Structural equivalence refers to a concept of vertex similarity in graph theory. Two vertices are considered structurally equivalent if they share connections with the same third-party vertices. By applying this equivalence relation, we can categorize vertices into several equivalence classes. In the context of multipartite graphs, each equivalence class is a subset of an independent set, representing a distinct perspective in our scenario. Consequently, outputs from the same perspective are demarcated into structural equivalence classes, of which outputs exhibit consistent behavior or possessing consistent meanings.

We define several intra-consistency measures based on the structural equivalence classes within each perspective. We denote the structural equivalence classes of $v_i$ as $S(v_i)$. The neighbors of $v_i$ can be partitioned into subsets $\{N_t(v_i)|t = 1, ...\}$ depending on the perspective they belong to.

- **Cardinality** utilizes the cardinality of the structural equivalence class as score.
- **Weight** is inspired by an intuition from the human discussion process that *Truth always rests with the majority*. The more agreements an idea receives, the higher the likelihood that it is correct. The neighbors of structural equivalence classes reach a consensus with them to some degree. Therefore, **Weight** multiply the cardinality of different neighbor subsets $\{N_t(v_i)|t = 1, ...\}$ to measure the agreements that the structural equivalence class receives.
- **Weighted Cardinality** combines both **Cardinality** and **Weight** by multiplication.

| Method | Expression |
|---|---|
| Bayes Risk | $\varphi(v_i) = C \cdot \sum_{v_j \in K(v_i)} \text{bleu}(v_i, v_j)$ |
| Cardinality | $\varphi(v_i) = C \cdot |S(v_i)|$ |
| Weight | $\varphi(v_i) = C \cdot \prod_t |N_t(v_i)|$ |
| Weighted Cardinality | $\varphi(v_i) = C \cdot |S(v_i)| \cdot \prod_t |N_t(v_i)|$ |
| Uniform | $\varphi(v_i) = C$ |
| Probability | $\varphi(v_i) = C \cdot \log p(v_i; \theta_{LLM})$ |
| Label | $\varphi(v_i) = \mathbf{1}_{v_i \text{ is label}}$ |

Table 1: Mathematical expressions of different intra-consistency measures. $C$ is the normalizing constant so that the measures of outputs within one perspective sum up to 1. $K(v_i)$ is the independent set of $v_i$ corresponding to its perspective. $S(v_i)$ is the structural equivalence class of $v_i$.

Considering the fact that the intra-consistency criteria $\mathcal{L}_{intra}$ simply utilizes $\boldsymbol{y}$ as a supervised signal without additional requirements on it, we can likewise introduce other prior information as $\boldsymbol{y}$ in addition to intra-consistency measures.

- **Uniform** is the baseline without any prior information and treats every vertex equally.
- **Probability** utilizes the probability of LLM generating the output sequence as score, which reflects the belief of the model about the output.
- **Label** introduces partial golden labels for supervision.

## 4 EXPERIMENT

### 4.1 EXPERIMENT SETTINGS

**Dataset and Metrics**    We conduct experiments on four widely used Python code generation benchmarks, including HumanEval, HumanEval+, MBPP and CodeContests. HumanEval (Chen et al., 2021) and MBPP (Austin et al., 2021) are two hand-written Python programming problems. HumanEval+ (Liu et al., 2023) adds more unit tests based on HumanEval. CodeContests (Li et al., 2022) is a much more challenging dataset consisting of competition problems from the Codeforces platform. Following Chen et al. (2021), we use Pass@$k$ for evaluation. It is an unbiased estimator of the probability that at least one out of the $k$ solutions generated by the model passes unit tests. Details about the metric can be found in Appendix C.

**Implementation and Baselines**    We compare several baselines from different LLMs for code like ChatGPT[3] (GPT-3.5-Turbo), GPT-4 (OpenAI, 2023), Code Llama (Touvron et al., 2023) and WizardCoder (Luo et al., 2023), to other approaches enhancing LLMs on code generation during inference, including Self-consistency (Wang et al., 2022), MBR-EXEC (Shi et al., 2022), CodeT (Chen et al., 2022) and Self-collaboration (Dong et al., 2023). For our framework, we employ GPT-3.5-Turbo as the foundation model to generate 200 solutions, 100 specifications and 500 test cases for each problem. For a fair comparison, we employ the same solutions and test cases if needed for other methods. For MPSC-Label, we use a human-written script[4] to extract test cases provided in docstrings as golden labels for the test case perspective. Further details regarding the implementation of our method and baselines are provided in Appendix E.

### 4.2 MAIN RESULTS

The experimental results on the four benchmarks are presented in Table 2. We observe that MPSC consistently enhances the code generation capabilities across all benchmarks with a remarkable margin of improvement. Particularly, when $k$ is set to 1, which is the most prevalent scenario in real-world applications, the performance improvement is notably significant (+17.6% on HumanEval, +17.61% on HumanEval+, +6.5% on MBPP and +11.82% on CodeContests). With the foundation model GPT-3.5-Turbo, our MPSC can even outperform GPT-4 in Pass@1 across all benchmarks

---

[3]https://chat.openai.com/

[4]Noted that MBPP doesn't provide test cases in docstrings.

| Benchmark | **HumanEval** | | | **HumanEval+** | | |
|---|---|---|---|---|---|---|
| Metric | Pass@1 | Pass@2 | Pass@5 | Pass@1 | Pass@2 | Pass@5 |
| GPT-3.5-Turbo | 68.38 | 76.24 | 83.15 | 58.75 | 66.58 | 73.96 |
| GPT-4 | 81.55 | 86.39 | **90.49** | 71.43 | 76.50 | **80.87** |
| Code Llama-Instruct | 62.20 | - | - | - | - | - |
| WizardCoder | 73.20 | - | - | - | - | - |
| Self-consistency | 73.86 | 73.93 | 74.10 | 63.50 | 64.70 | 65.67 |
| MBR-EXEC | 72.96 | 76.47 | 79.00 | 62.12 | 67.08 | 71.38 |
| CodeT | 78.05 | 78.05 | 78.30 | 67.87 | 68.75 | 69.65 |
| Self-collaboration | 74.40 | - | - | - | - | - |
| MPSC-Uniform | $82.32_{+13.94}$ | $83.86_{+7.62}$ | $84.79_{+1.64}$ | $71.29_{+12.54}$ | $74.32_{+7.74}$ | $76.17_{+2.21}$ |
| MPSC-Bayes Risk | $83.54_{+15.16}$ | $84.76_{+8.52}$ | $85.98_{+2.83}$ | $73.78_{+15.03}$ | $75.61_{+9.03}$ | $76.83_{+2.87}$ |
| MPSC-Cardinality | $84.15_{+15.77}$ | $84.27_{+8.03}$ | $85.06_{+1.91}$ | $74.63_{+15.88}$ | $75.16_{+8.58}$ | $76.73_{+2.77}$ |
| MPSC-Weight | $83.69_{+15.31}$ | $84.56_{+8.32}$ | $86.23_{+3.08}$ | $74.10_{+15.35}$ | $74.97_{+8.39}$ | $77.97_{+4.01}$ |
| MPSC-Weighted Cardinality | **$85.98_{+17.60}$** | $85.48_{+9.24}$ | $86.25_{+3.10}$ | **$76.36_{+17.61}$** | **$76.77_{+10.19}$** | $77.59_{+3.63}$ |
| MPSC-Label | $85.37_{+16.99}$ | **$86.60_{+10.36}$** | $86.35_{+3.20}$ | $74.95_{+16.20}$ | $76.60_{+10.02}$ | $76.96_{+3.00}$ |
| Benchmark | **MBPP** | | | **CodeContests** | | |
| Metric | Pass@1 | Pass@2 | Pass@5 | Pass@1 | Pass@2 | Pass@5 |
| GPT-3.5-Turbo | 66.80 | 72.34 | 76.60 | 2.57 | 4.22 | 7.16 |
| GPT-4 | 71.26 | **74.27** | **76.99** | 6.07 | 8.23 | 11.67 |
| Code Llama-Instruct | 61.20 | - | - | - | - | - |
| WizardCoder | 61.20 | - | - | 2.15 | 3.40 | 5.37 |
| Self-consistency | 71.70 | 71.73 | 71.82 | 8.10 | 8.42 | 8.48 |
| MBR-EXEC | 70.79 | 73.14 | 74.85 | 8.25 | 8.87 | 9.08 |
| CodeT | 71.90 | 71.95 | 72.02 | 9.92 | 10.18 | 10.30 |
| Self-collaboration | 68.20 | - | - | - | - | - |
| MPSC-Uniform | $68.44_{+1.64}$ | $70.21_{-2.13}$ | $71.61_{-4.99}$ | $5.45_{+2.88}$ | $6.05_{+1.83}$ | $6.76_{-0.40}$ |
| MPSC-Bayes Risk | **$73.30_{+6.50}$** | $73.54_{+1.20}$ | $73.77_{-2.83}$ | $8.48_{+5.91}$ | $9.70_{+5.48}$ | $10.30_{+3.14}$ |
| MPSC-Cardinality | $72.13_{+5.33}$ | $72.14_{-0.20}$ | $72.15_{-4.45}$ | $8.84_{+6.27}$ | $9.69_{+5.47}$ | $9.70_{+2.54}$ |
| MPSC-Weight | $71.90_{+5.10}$ | $72.62_{+0.28}$ | $72.72_{-3.88}$ | $6.03_{+3.46}$ | $6.66_{+2.44}$ | $8.59_{+1.43}$ |
| MPSC-Weighted Cardinality | $73.17_{+6.37}$ | $73.24_{+0.90}$ | $73.31_{-3.29}$ | $11.26_{+8.69}$ | $11.51_{+7.29}$ | $12.12_{+4.96}$ |
| MPSC-Label | - | - | - | **$14.39_{+11.82}$** | **$17.16_{+12.94}$** | **$17.76_{+10.6}$** |

Table 2: The results on four code generation benchmarks. The foundation model for MPSC, Self-consistency, MBR-EXEC, CodeT, Self-collaboration are all GPT-3.5-Turbo. The improvements are calculated between our method and GPT-3.5-Turbo. The best and second best performance for each dataset are shown in **bold** and underline.

(+4.43% on HumanEval, +4.93% on HumanEval+, +2.04% on MBPP and +8.32% on CodeContests). Compared to other approaches which also enhance LLMs, our MPSC still shows consistent advantages in all benchmarks, excluding the Pass@5 score in MBPP benchmark. The strong performance of MPSC-Uniform demonstrates that relying solely on inter-consistency can also boost coding ability of LLM. On the other hand, incorporating various types of intra-consistency information leads to even greater improvements. Specifically, MPSC-Label and MPSC-Weighted Cardinality exhibit particularly strong results. Surprisingly, MPSC-Weighted Cardinality can match or even surpass MPSC-Label, which leverages the external supervision signals from golden test cases in docstrings. This observation further highlights the significance of consistency information in LLMs.

### 4.3 FURTHER ANALYSIS

**Ablation Study**  We conduct an ablation study to examine the impact of different perspectives on MPSC. We adopt Uniform as the intra-consistency measure in this experiment. The results are presented in Table 4. Evidently, both the specification and test case perspectives play crucial roles in our framework. Additionally, the results indicate that test cases contribute more to the improvements than specifications. We attribute the observation to the superior quality of test cases, as generating an accurate test case is considerably simpler than abstracting a comprehensive and sound specification.

**Generalization over different LLMs**  MPSC is a model-agnostic framework that assumes black-box access to the underlying foundation model. To examine the generalization of MPSC, we employ open-source LLMs, in addition to ChatGPT, as foundation models. In specific, we consider

| Benchmark | HumanEval | | | HumanEval+ | | |
|---|---|---|---|---|---|---|
| Metric | Pass@1 | Pass@2 | Pass@5 | Pass@1 | Pass@2 | Pass@5 |
| WizzardCoder-34B[†] | 67.84 | 72.12 | 75.98 | 58.70 | 62.88 | 66.88 |
| +MPSC | $79.27_{+11.43}$ | $78.10_{+5.98}$ | $78.28_{+2.30}$ | $68.67_{+9.97}$ | $68.43_{+5.55}$ | $68.52_{+1.64}$ |
| Code Llama-34B | 51.78 | 59.24 | 67.07 | 41.49 | 48.30 | 55.93 |
| +MPSC | $73.78_{+22.00}$ | $75.14_{+15.90}$ | $77.17_{+10.10}$ | $59.76_{+18.27}$ | $61.11_{+12.81}$ | $61.92_{+5.99}$ |
| WizzardCoder-13B | 60.35 | 66.10 | 72.01 | 50.25 | 56.00 | 61.98 |
| +MPSC | $75.00_{+14.65}$ | $76.83_{+10.73}$ | $76.44_{+4.43}$ | $62.80_{+12.55}$ | $64.02_{+8.02}$ | $64.85_{+2.87}$ |
| Code Llama-13B | 44.63 | 50.99 | 57.86 | 35.93 | 41.71 | 48.19 |
| +MPSC | $67.07_{+22.44}$ | $68.29_{+17.30}$ | $69.36_{+11.50}$ | $53.66_{+17.73}$ | $54.27_{+12.56}$ | $54.73_{+6.54}$ |
| WizzardCoder-7B | 53.81 | 59.62 | 66.06 | 45.06 | 50.83 | 57.69 |
| +MPSC | $70.52_{+16.71}$ | $69.90_{+10.28}$ | $70.85_{+4.79}$ | $59.62_{+14.56}$ | $60.95_{+10.12}$ | $63.01_{+5.32}$ |
| Code Llama-7B | 39.38 | 45.18 | 52.79 | 34.33 | 39.18 | 45.25 |
| +MPSC | $62.20_{+22.82}$ | $62.33_{+17.15}$ | $63.14_{+10.35}$ | $52.44_{+18.11}$ | $53.66_{+14.48}$ | $54.24_{+8.99}$ |

Table 3: The performance of MPSC with different foundation models. [†]: We use nucleus sampling with temperature as 0.2 instead of greedy generation in this experiment.

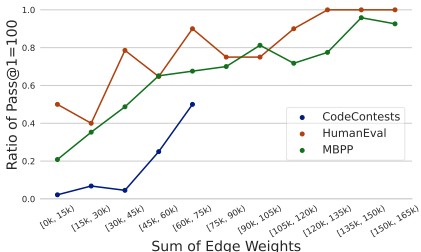

Figure 3: The correlation between the performance of MPSC and the edge density.

| Method | Pass@1 | Pass@2 | Pass@5 |
|---|---|---|---|
| Ours | 82.32 | 83.86 | 84.79 |
| w/o Specification | 79.27 | 81.42 | 83.57 |
| w/o Test case | 75.36 | 78.23 | 79.57 |
| w/o All | 68.38 | 76.24 | 83.15 |

Table 4: The ablation study results on HumanEval. The measure of inter-consistency is Uniform.

two highly proficient LLMs for code, Code Llama and WizardCoder in Python with three scales of parameters, 7B, 13B and 34B. The experiment results presented in Table 3 demonstrate that MPSC consistently yields significant improvements across all models, which proves the extraordinary generalization of our proposed framework.

**Impact of Edge Sparsity**   Our framework significantly depends on the inter-consistency information between model outputs, which is represented as edges within the constructed graph. A critical question that arises concerns the impact of edge sparsity on the framework's efficacy. To address this, we categorize all queries in the dataset into distinct bins based on the total edge weights in their corresponding graphs and compute the perfect performance ratio (i.e., Pass@1=100) for each bin. In this experiment, we employ the MPSC-Uniform configuration[5]. Figure 3 illustrates the correlation between edge density and performance. The results clearly demonstrate a positive correlation between the number of edges and the overall performance of our framework.

| #Test cases | #Specification | | | |
|---|---|---|---|---|
| | 10 | 20 | 50 | 100 |
| 50 | 84.37 | 83.85 | 83.85 | 82.20 |
| 100 | 84.69 | 85.30 | 85.30 | 82.86 |
| 200 | 84.10 | 84.71 | 85.93 | 84.71 |
| 500 | 84.76 | 85.37 | 85.37 | 85.98 |

Table 5: Pass@1 of MPSC with different sampling numbers on HumanEval.

| #Test cases | #Specification | | | |
|---|---|---|---|---|
| | 10 | 20 | 50 | 100 |
| 50 | 71.23 | 72.34 | 72.33 | 72.96 |
| 100 | 71.69 | 72.16 | 72.40 | 72.70 |
| 200 | 71.69 | 72.39 | 72.40 | 72.46 |
| 500 | 70.58 | 70.66 | 71.83 | 72.94 |

Table 6: Pass@1 of MPSC with different sampling numbers on MBPP.

---

[5]The Uniform configuration is utilized to eliminate the influence of intra-consistency information.

**Impact of Sampling Number**    The sampling number of different perspectives may also affect the performance of MPSC. To examine the effect, we conduct an analysis experiment by reducing the number of specifications and test cases. We use the MPSC-Weighted Cardinality configuration[6] in this experiment. As shown in Table 5 and 6, MPSC generally suffers a slight degradation in performance when fewer specifications or test cases are used, which is consistent with our intuition. However, the performance decline is relatively small (1.6% for HumanEval and 1.7% for MBPP) with only 10% of specifications and test cases are retained. The observation highlights the remarkable performance and efficiency of MPSC, suggesting the potential for real-world application with reduced computational costs.

## 5    RELATED WORK

**Prompting Techniques on Consistency**    Based on Chain-of-thought mechanism Wei et al. (2022), many previous works have adopted various prompting techniques and decoding strategies to reveal the consistency of LLM outputs and further enhance the capabilities of LLMs. One line of approaches decodes multiple times from the same perspective and aggregate the results (Wang et al., 2022; Zhou et al., 2022; Jung et al., 2022; Sun et al., 2022). For example, Wang et al. (2022) targets tasks with fixed answer sets and scores each answer based on the output frequency. Building on this, Sun et al. (2022) introduces recitation as context for augmentation. Jung et al. (2022) is also a post-hoc method to enhance the reasoning abilities of LLM. They focus on the two-value entailment relations (True or False) between statements and explanations. They treat the inference process as a weighted MAX-SAT problem and utilize a logistic solver to solve it. Another line draws inspiration from the "Dual Process" in cognitive science, which posits that human reasoning is dominated by System 1 and System 2 (Daniel, 2017; Sloman, 1996). As a result, these approaches require LLMs to play different roles like generator (i.e., System 1) and verifier (i.e., System 2), and optimize the result iteratively by a conversational way (Madaan et al., 2023; Shinn et al., 2023; Zhu et al., 2023). Xiong et al. (2023) also proposes the concept of "inter-consistency". Instead of referring to the consistency within the same LLM, they focus to tackle the inter-inconsistency problem between different models with a formal debate framework.

**LLM for Code**    LLMs pretrained on large-scale code data have demonstrated strong capabilities in the field of code generation (Chen et al., 2021; Li et al., 2022; Nijkamp et al., 2023; Fried et al., 2022; Li et al., 2023; Rozière et al., 2023; Luo et al., 2023). However, they remain unreliable, particularly in scenarios involving complex input-output mappings. Several methods have been proposed to mitigate the phenomenon (Shi et al., 2022; Chen et al., 2022; Zhang et al., 2022; Key et al., 2022; Ni et al., 2023; Dong et al., 2023; Olausson et al., 2023; Chen et al., 2023; Zhang et al., 2023). For example, Shi et al. (2022) matches the execution results of generated solutions for minimum Bayes risk selection. CodeT (Chen et al., 2022) additionally generates test cases to verify the generated solutions. LEVER Ni et al. (2023) learns a verifier to predict the correctness of the program based on the NL, program and execution results.

**Ranking on Graph**    In our framework, the final problem is abstracted and transformed into a ranking problem in graph. There exists some renowned graph ranking algorithms like PageRank (Page et al., 1998) and HITS (Kleinberg, 1999). While our approach is inspired by manifold ranking (Zhou et al., 2003b), which is built upon a regularization framework on discrete spaces (i.e. graphs in this scenario) (Zhou et al., 2003a; Zhou & Schölkopf, 2004; 2005).

## 6    FUTURE WORK AND CONCLUSION

In this paper, we present a novel Large Language Model (LLM) decoding strategy, Multi-Perspective Self-Consistency (MPSC), aimed at enhancing the performance of LLMs in complex reasoning tasks where a single attempt may not guarantee the accuracy of the output. The proposed MPSC strategy capitalizes on both intra- and inter-consistency among the generated outputs from multiple perspectives to identify the most reliable answer. We conduct comprehensive experiments in code generation. Evaluation results demonstrate that MPSC achieves the state-of-the-art performance in

---

[6]We use Weighted Cardinality since sampling numbers affect both inter- and intra-consistency information.

four benchmarks. Since MPSC framework is model-agnostic and task-agnostic, one can also apply MPSC to other textual generation tasks like math problem solving and question answering. However, unlike code generation, where code interpreter can measure the agreement between outputs in a deterministic way, assessing the agreement between natural language outputs may not be easy. So we leave it as future work to apply MPSC in other scenarios.

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

## A    DETAILS OF THE ITERATIVE ALGORITHM

**Description of Algorithm**    The iterative algorithm is shown in Algorithm 1.

---
**Algorithm 1:** Iterative Optimization

---
**Input:** degree matrix $D = \mathrm{diag}(d_1, ..., d_N)$, initialization score vector $y$, weighted adjacency matrix $W$, threshold $\epsilon$
**Output:** optimal confidence score vector $f^*$
**begin**
$\quad f^{(0)} \longleftarrow y$
$\quad T \longleftarrow D^{-\frac{1}{2}} W D^{-\frac{1}{2}}$
$\quad i \longleftarrow 0$
$\quad$ **Do**
$\quad\quad f^{(i+1)} \longleftarrow \alpha T f^{(i)} + (1 - \alpha) y$
$\quad\quad i \longleftarrow i + 1$
$\quad$ **While** $||f^{(i)} - f^{(i-1)}|| \le \epsilon$
$\quad f^* \longleftarrow f^{(i)}$
$\quad$ **return** $f^*$

---

**Proof of Convergence**    We first expand the expression of $f^{(n)}$ according to the recursive formula

$$f^{(n)} = \alpha T f^{(n-1)} + (1 - \alpha) y$$

$$= (\alpha T)^{n-1} f^{(0)} + (1 - \alpha) \sum_{i=0}^{n-1} (\alpha T)^i y$$

Notice that $T$ is similar to a stochastic matrix $W D^{-1} = D^{\frac{1}{2}} (D^{-\frac{1}{2}} W D^{-\frac{1}{2}}) D^{-\frac{1}{2}} = D^{\frac{1}{2}} T D^{-\frac{1}{2}}$. Therefore the eigenvalues of $\alpha T$ are in $[-\alpha, \alpha]$. With $\alpha \in (0, 1)$, we have

$$\lim_{n \to \infty} (\alpha T)^n = 0$$

$$\lim_{n \to \infty} \sum_{i=0}^{n-1} (\alpha T)^i = (I - \alpha T)^{-1}$$

Therefore

$$f^* = \lim_{n \to \infty} f^{(n)} = (1 - \alpha)(1 - \alpha T)^{-1} y$$

| Benchmark | **HumanEval** | | | **HumanEval+** | | |
|---|---|---|---|---|---|---|
| Metric | Pass@1 | Pass@2 | Pass@5 | Pass@1 | Pass@2 | Pass@5 |
| MPSC-Uniform | 82.32/82.32 | 83.86/84.18 | 84.79/84.83 | 71.29/71.29 | 74.32/74.83 | 76.17/76.17 |
| MPSC-Bayes | 83.54/83.54 | 84.76/84.76 | 85.98/85.98 | 73.78/73.78 | 75.61/75.61 | 76.83/76.83 |
| MPSC-Cardinality | 84.15/84.15 | 84.27/84.28 | 85.06/85.09 | 74.63/74.62 | 75.16/75.89 | 76.73/77.46 |
| MPSC-Weight | 83.69/83.69 | 84.56/85.06 | 86.23/86.31 | 74.10/74.22 | 74.97/76.04 | 77.97/79.24 |
| MPSC-Weighted Cardinality | 85.98/85.98 | 85.48/85.77 | 86.25/86.59 | 76.36/76.25 | 76.77/77.13 | 77.59/77.75 |
| Benchmark | **MBPP** | | | **CodeContests** | | |
| Metric | Pass@1 | Pass@2 | Pass@5 | Pass@1 | Pass@2 | Pass@5 |
| MPSC-Uniform | 68.44/68.50 | 70.21/70.35 | 71.61/72.04 | 5.45/5.43 | 6.05/6.02 | 6.76/6.79 |
| MPSC-Bayes | 73.30/73.30 | 73.54/73.54 | 73.77/73.77 | 8.48/8.48 | 9.70/9.70 | 10.30/10.30 |
| MPSC-Cardinality | 72.13/72.15 | 72.14/72.17 | 72.15/72.22 | 8.84/8.89 | 9.69/9.66 | 9.70/9.70 |
| MPSC-Weight | 71.90/71.76 | 72.62/72.84 | 72.72/73.02 | 6.03/6.03 | 6.66/6.67 | 8.59/8.86 |
| MPSC-Weighted Cardinality | 73.17/73.21 | 73.24/73.31 | 73.31/73.49 | 11.26/10.30 | 11.51/11.52 | 12.12/12.12 |

Table 7: Performance of MPSC optimized by the iterative algorithm or calculating closed-form solution directly. The results are presented in form of `(iterative algorithm / closed-form solution)`.

**Proof of Equivalence** Denote the optimization function as

$$
\mathcal{F} = \alpha \boldsymbol{f}^T (\boldsymbol{I} - \boldsymbol{D}^{-\frac{1}{2}} \boldsymbol{W} \boldsymbol{D}^{\frac{1}{2}}) \boldsymbol{f} + \frac{(1-\alpha)}{2} (\boldsymbol{f} - \boldsymbol{y})^2
$$
$$
= \alpha \boldsymbol{f}^T (\boldsymbol{I} - \boldsymbol{T}) \boldsymbol{f} + \frac{(1-\alpha)}{2} (\boldsymbol{f} - \boldsymbol{y})^2
$$

Differentiate $\mathcal{F}$ with respect to $\boldsymbol{f}$, we have

$$
\frac{\partial \mathcal{F}}{\partial \boldsymbol{f}} = \alpha (\boldsymbol{I} - \boldsymbol{T}) \boldsymbol{f} + (1-\alpha)(\boldsymbol{f} - \boldsymbol{y})
$$

Let the derivatives to 0, the solution $\boldsymbol{f}' = (1-\alpha)(\boldsymbol{I} - \alpha \boldsymbol{T})^{-1} \boldsymbol{y} = \boldsymbol{f}^*$. Therefore, the iterative algorithm is actually optimizing the objective function.

**Results of Closed-form Solution** Despite the existence of a closed-form solution for the optimization problem, the required matrix inversion operation is computationally expensive. Conversely, the iterative algorithm exhibits rapid convergence and demonstrates strong empirical performance. Consequently, we employ the iterative algorithm in our experiments. Additionally, we provide the performance of the closed-form solution in Table 7. Our results indicate that the iterative algorithm achieves a performance on par with that of the direct computation of the closed-form solution.

## B  IMPLEMENTATION OF INTER-CONSISTENCY

We present the code snippets measuring the inter-consistency between each pair of perspectives in Listing 1, 2, 3. After execution with Python interpreter, the `final_result` is acquired as $\omega(v_i, v_j)$.

```python
"""Generated specifications"""
# Pre-conditions
def preconditions(input):
...

# Post-conditions
def postconditions(input, output):
...

"""Generated test cases"""
test_case = {'input': ...,
             'output': ...}

```

```
14  """Check inter-consistency"""
15  def check():
16      pass_result = None
17      try:
18          preconditions(test_case['input'])
19          postconditions(test_case['input'], test_case['output'])
20          pass_result = True
21      except Exception as e:
22          pass_result = False
23      return pass_result
24  global final_result
25  final_result = check()
```

Listing 1: Inter-consistency between specifications and test cases.

```
1   """Generated solutions"""
2   def entry_point(input):
3   ...
4
5   """Generated specifications"""
6   # Pre-conditions
7   def preconditions(input):
8   ...
9
10  # Post-conditions
11  def postconditions(input, output):
12  ...
13
14  """Generated casual inputs"""
15  casual_inputs = [...]
16
17  """Check inter-consistency"""
18  def check():
19      pass_result = []
20      for ci in casual_inputs:
21          try:
22              output = entry_point(ci)
23              postconditions(ci, output)
24              pass_result.append(True)
25          except Exception as e:
26              pass_result.append(False)
27      return sum(pass_result) / len(pass_result)
28  global final_result
29  final_result = check()
```

Listing 2: Inter-consistency between solutions and specifications.

```
1   """Generated solutions"""
2   def entry_point(input):
3   ...
4
5   """Generated test cases"""
6   test_case = {'input': ...,
7               'output': ...}
8
9   """Check inter-consistency"""
10  def check():
11      try:
12          output = entry_point(test_case['input'])
13          pass_result = (output == test_case['output'])
14      except Exception as e:
15          pass_result = False
16      return pass_result
17  global final_result
```

```
18  final_result = check()
```
Listing 3: Inter-consistency between solutions and test cases.

## C  DISCUSSION ABOUT PASS@$k$

In this section, we discuss the flaws of Pass@$k$ and propose a variant for evaluating methods involved selection and filtering.

Chen et al. (2021) propose an unbiased estimator called Pass@$k$, which estimates the probability of a model passing unit tests within $k$ attempts. In specific, Chen et al. (2021) first samples a total of $n$ solutions with $c$ of them are correct, randomly samples $k$ solutions for testing, and use the probability of passing tests for estimation,

$$\text{Pass@}k = 1 - \frac{\binom{n-c}{k}}{\binom{n}{k}}$$

Although their implementation serves as an effective measure of the code generation ability of different foundation models (referred to as the first category of methods in the following), it is not suitable to evaluate methods involving filtering or selection during the inference stage (Li et al., 2022; Chen et al., 2022) (referred to as the second category of methods in the following), as the $n$ generated solutions are identical.

To address the limitation, we implement a variant of Pass@$k$. We assume each method provides a score function over the $n$ generated solutions, which provides a unified view for the two method categories and hence enables a fair comparison. Based on the original definition of Pass@$k$, we evaluate the method by testing the top-$k$ solutions with the highest scores. As the score function may assign the same score to multiple solutions, the test result of the top-k is not deterministic but an expected value.

Mathematically, let's assume that a method sequentially arranges outputs into an ordered list $\{s_1, ..., s_n\}$, such that $\forall i > j, s_i \preceq s_j$ according to their scores. We define a set of solutions $\mathbb{S}^k = \{s_i | s_k \preceq s_i\}$, which represents the solution set selected by the method. Suppose the cardinality of $\mathbb{S}^k$ is $\hat{n}$, the number of correct solutions within $\mathbb{S}^k$ is $\hat{c}$. Noted that $\hat{n} \geq k$, and thus we uniformly sample $k$ solutions $\{s'_1, ..., s'_k\}$ from $\mathbb{S}^k$ for estimation,

$$\begin{aligned}
\text{Pass@}k \text{ (Ours)} &= \mathbb{E}_{s'_1, ..., s'_k}\left[\mathbf{1}_{\cup_{i=1}^k s'_i \text{ is correct}}\right] \\
&= \Pr(\cup_{i=1}^k s'_i \text{ is correct}) \\
&= 1 - \Pr(\cap_{i=1}^k s'_i \text{ is incorrect}) \\
&= 1 - \frac{\binom{\hat{n}-\hat{c}}{k}}{\binom{\hat{n}}{k}}
\end{aligned}$$

For a the first category of methods, $\hat{n}$ equals $n$ since it treats each solution equally. As a result, our implementation of Pass@$k$ is the same as the original implementation in Chen et al. (2021).

Chen et al. (2022) also implements another variant of Pass@$k$ with a rolling solution selection. We present the performance of MPSC evaluated by their implementation as a supplement in Table 8.

## D  OTHER ANALYSIS

**Incorporating Golden Test Cases**  In practical applications of code generation, users often provide a limited number of test cases to outline the desired functionality, thereby assisting the model in generating code that aligns with the requirements. In this study, we investigate the potential performance improvements of the MPSC model in such scenarios by incorporating various quantities of golden test cases. These golden test cases are generated and then validated using canonical solutions provided in the benchmarks. We assess the performance of the MPSC-Label model on the HumanEval dataset and present the results in Table 9. The substantial performance enhancements achieved with the inclusion of merely five golden test cases underscore the feasibility of implementing MPSC in user-interactive application scenarios.

| Benchmark | HumanEval | | | HumanEval+ | | |
|---|---|---|---|---|---|---|
| Method | Pass@1 | Pass@2 | Pass@5 | Pass@1 | Pass@2 | Pass@5 |
| GPT-3.5-Turbo | 68.38 | 76.24 | 83.15 | 58.75 | 66.58 | 73.96 |
| CodeT | 78.05 | 85.98 | 92.06 | 67.87 | 79.12 | 83.87 |
| MPSC-Uniform | 81.71 | 87.20 | 89.63 | 71.29 | 79.51 | 82.98 |
| MPSC-Bayes Risk | 83.54 | 84.76 | 85.98 | 73.78 | 75.61 | 76.83 |
| MPSC-Cardinality | 84.15 | 87.80 | 90.85 | 74.63 | 81.15 | 84.11 |
| MPSC-Weight | 83.69 | 86.13 | 91.62 | 74.10 | 78.29 | 84.71 |
| MPSC-Weighted Cardinality | 85.98 | 87.20 | 92.07 | 76.36 | 80.50 | 85.33 |
| MPSC-Label | 85.37 | 89.02 | 90.85 | 74.95 | 80.83 | 83.89 |
| Benchmark | MBPP | | | CodeContests | | |
| Method | Pass@1 | Pass@2 | Pass@5 | Pass@1 | Pass@2 | Pass@5 |
| GPT-3.5-Turbo | 66.80 | 72.34 | 76.60 | 2.57 | 4.22 | 7.16 |
| CodeT | 71.90 | 76.82 | 80.35 | 9.92 | 11.43 | 14.46 |
| MPSC-Uniform | 68.44 | 73.30 | 79.63 | 5.43 | 5.45 | 8.43 |
| MPSC-Bayes Risk | 73.30 | 73.54 | 73.77 | 8.48 | 9.70 | 10.30 |
| MPSC-Cardinality | 72.13 | 77.05 | 80.37 | 8.84 | 11.32 | 13.19 |
| MPSC-Weight | 71.90 | 74.17 | 79.38 | 6.03 | 7.19 | 9.04 |
| MPSC-Weighted Cardinality | 73.17 | 76.28 | 79.85 | 11.26 | 11.36 | 15.59 |
| MPSC-Label | - | - | - | 12.60 | 17.37 | 18.73 |

Table 8: Performance of MPSC evaluated by the Pass@$k$ metric implemented by Chen et al. (2022).

| Method | Pass@1 | Pass@2 | Pass@5 |
|---|---|---|---|
| MPSC | 82.32 | 83.86 | 84.79 |
| + 1 test case | 85.37 | 86.59 | 85.13 |
| + 2 test cases | 85.98 | 86.18 | 85.36 |
| + 5 test cases | 88.41 | 89.23 | 88.69 |
| + 10 test cases | 89.02 | 90.24 | 88.81 |

Table 9: Performance of MPSC with different numbers of golden test cases provided by users on HumanEval.

**Analysis of Other Perspectives**   MPSC not only selects the optimal output from the target perspective but also chooses outputs from auxiliary perspectives, thereby generating corresponding by-products. In the context of code generation, which is the primary focus of this paper, MPSC can additionally identify more reliable test cases and specifications. We evaluate the quality of these by-products and present the results in Table 10. The experimental results demonstrate that MPSC is proficient in selecting high-quality outputs across all relevant perspectives.

| Benchmark | HumanEval | | | MBPP | | |
|---|---|---|---|---|---|---|
| Metric | Pass@1 | Pass@2 | Pass@5 | Pass@1 | Pass@2 | Pass@5 |
| Specification | | | | | | |
| GPT-3.5-Turbo | 45.66 | 58.56 | 72.07 | 50.83 | 60.35 | 69.15 |
| MPSC | 76.10 | 77.16 | 78.43 | 69.98 | 72.30 | 74.51 |
| Test case | | | | | | |
| GPT-3.5-Turbo | 63.83 | 80.71 | 93.23 | 24.54 | 30.97 | 36.43 |
| MPSC | 98.17 | 98.17 | 98.17 | 37.06 | 37.53 | 38.39 |

Table 10: The quality of specifications and test cases selected by MPSC.

**Effect of Alpha**   We explore the impact of the weight parameter $\alpha$ on the performance of the proposed MPSC framework. $\alpha$ serves as a balancing factor between the inter-consistency and intra-consistency information. We conduct experiments to investigate the relationship between the value of $\alpha$ and the performance of MPSC. The experimental results are presented in Figure 4. The results reveal that there is no evident correlation between the performance of MPSC and the choice of $\alpha$.

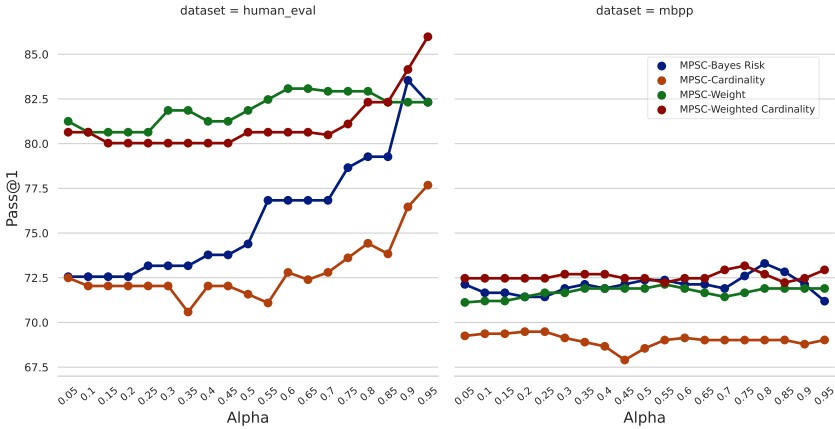

Figure 4: Performance of MPSC with different values of alpha.

This observation suggests that neither inter-consistency nor intra-consistency information exhibits a distinct advantage or disadvantage over the other, and the optimal selection of $\alpha$ is contingent upon the specific application scenario.

**Stability of MPSC**   We explore the stability of MPSC with respect to the sampling process. We conduct the sampling process of WizardCoder with three random seeds and then assess the performance of MPSC on the generated solutions. The average results are shown in Table 11. It is evident that the improvement brought by MPSC is very stable.

| Benchmark | **HumanEval** | | | **HumanEval+** | | |
|---|---|---|---|---|---|---|
| Metric | Pass@1 | Pass@2 | Pass@5 | Pass@1 | Pass@2 | Pass@5 |
| WizzardCoder-34B | $66.04_{\pm1.27}$ | $70.95_{\pm0.84}$ | $75.51_{\pm0.43}$ | $56.12_{\pm1.83}$ | $60.54_{\pm1.66}$ | $64.67_{\pm1.56}$ |
| +MPSC | $79.8_{\pm1.31}$ | $80.32_{\pm1.88}$ | $80.56_{\pm2.58}$ | $66.04_{\pm2.55}$ | $66.63_{\pm1.39}$ | $67.37_{\pm1.58}$ |
| WizzardCoder-13B | $58.62_{\pm1.23}$ | $65.05_{\pm0.75}$ | $71.81_{\pm0.14}$ | $49.42_{\pm0.59}$ | $54.91_{\pm0.77}$ | $60.81_{\pm0.83}$ |
| +MPSC | $75.5_{\pm0.36}$ | $76.52_{\pm0.9}$ | $76.42_{\pm0.71}$ | $63.75_{\pm0.72}$ | $65.16_{\pm1.58}$ | $65.31_{\pm1.31}$ |
| WizzardCoder-7B | $52.2_{2.17}$ | $58.16_{\pm1.94}$ | $64.73_{\pm1.68}$ | $43.74_{\pm2.34}$ | $49.48_{\pm2.37}$ | $56.26_{\pm2.3}$ |
| +MPSC | $70.98_{\pm0.95}$ | $70.17_{\pm1.2}$ | $70.27_{\pm1.83}$ | $59.69_{\pm1.68}$ | $60.71_{\pm1.98}$ | $61.59_{\pm2.77}$ |

Table 11: The average performance of MPSC with three sample sets under different random seeds.

## E   EXPERIMENT SETTINGS AND BASELINES

**Baselines**   We incorporate various baselines in code generation. First of all, we include many strong large language models like ChatGPT(gpt-3.5-turbo 0315 version), GPT-4(gpt4-0515 version), Code Llama-Instruct-34B and WizardCoder-Python-34B. The specific hyper-parameters for inference of ChatGPT and GPT4 are shown in Table 12. We use the public released performance of WizardCoder and Code Llama directly.

| | |
|---|---|
| Temperature | 0.8 |
| Top P | 0.95 |
| Frequency Penalty | 0 |
| Presence Penalty | 0 |

Table 12: The Inference hyper-parameters of LLMs.

We also include several baselines like Self-Consistency MBR-EXEC, CodeT and Self-collaboration, which enhance the inference capability of LLMs.

- Self-Consistency: We implement this baseline following Chen et al. (2022). If two solution pass the same set of generated test cases and specifications, we regard them "consistent". Then we take a majority voting to rank solutions following Wang et al. (2022).
- MBR-EXEC: This baseline ranks solutions by minimum Bayes risk decoding based on the execution results in the generated test cases.
- CodeT: This baseline first uses generated test cases to verify each solution by code execution. Then it utilizes RANSAC algorithm to create consensus sets based on execution results. The size of consensus set is then used to rank solutions.

For a fair comparison with our proposed MPSC, we employ the same solutions generated by Chat-GPT for them to rerank. Since CodeT leverages generated test cases either, we use the same test cases generated by ChatGPT for both CodeT and MPSC.

For all baselines and MPSC, we sample 200 solutions following the conventional setting. We sample 500 test cases for MPSC and CodeT, which is the original setting in CodeT. In adition, we sample 100 specifications for MPSC. We employ our implementation of Pass@$k$ for all experiments.

**Prompt for MPSC**   We present the prompt to generate solutions, specifications and test cases in Table 13, 14, 15.

```Python
I want you to act like a Python programmer. I will give you the declaration of a function and comments about its property. You need to implement the body of the function in the code block. Do not modify any code I provide. Do not provide any explanations.

Here is the question.
```Python
{Docstrings}
```
```

Table 13: Prompt for generating solutions in the zero-shot manner.

```Python
# Given a docstring, continue to write the following code with 10 valid assertion statements to check the correctness of the function. Provide diverse test cases.
{Docstrings}
    pass

# check the correctness of with 10 different valid assertion statements in the form of "assert {entry point}(...) == ..."
assert
```

Table 14: Prompt for generating test cases in the zero-shot manner.

## F   AN EXAMPLE OF CONSTRUCTED GRAPH

Here we provide an example of constructed graph for one code generation query in MBPP dataset in Figure 5.

I want you to act as a python programmer. Given a docstring about a python method, you need to write its pre-conditions in one test function "def preconditions(input)" and post-conditions in another test function "def postconditions(input, output):". You should ensure invalid input or output of the method will raise error in the two test functions.
```Python
{Demonstration Docstrings 1}
    pass
#Pre-conditions
{Demonstration Preconditions 1}
#Post-conditions
{Demonstration Postconditions 1}
```

```Python
{Demonstration Docstrings 2}
    pass
#Pre-conditions
{Demonstration Preconditions 2}
#Post-conditions
{Demonstration Postconditions 2}
```

```Python
{Docstrings}
    pass

Table 15: Prompt for generating specifications with two demonstrations.

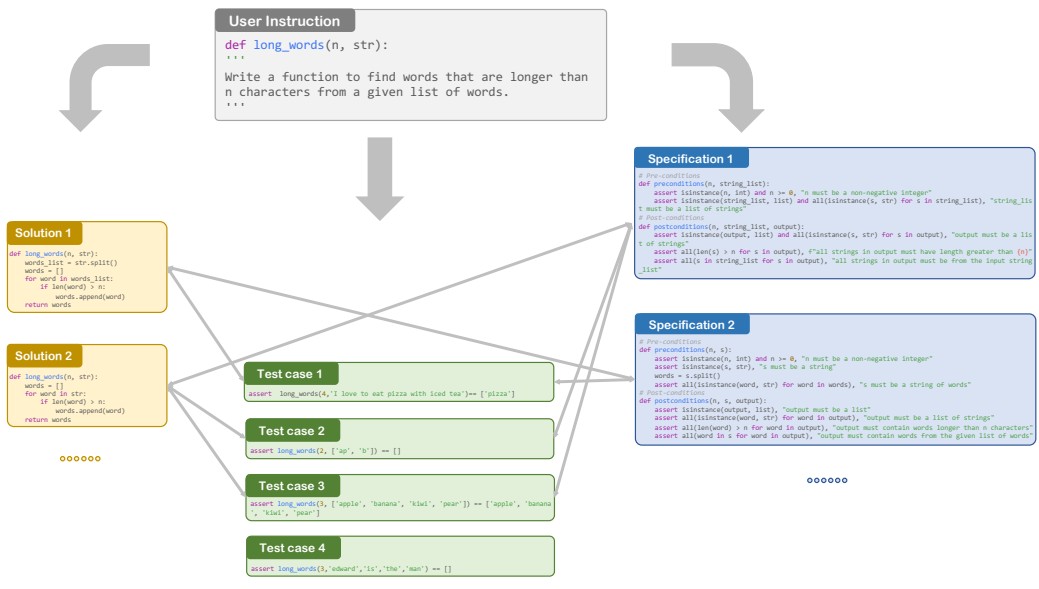

Figure 5: An example of a constructed graph is presented in this figure. For the sake of clarity, we only show a sub-graph featuring two solution vertices, two specification vertices, and four test case vertices. These vertices are categorically divided into two groups, each representing distinct interpretations of the user instruction. *Solution 1*, *Specification 2*, and *Test case 1* interpret the input parameter str as a string, while the others consider it as a list of strings. Consequently, edges only exist within each group. Notably, *Test case 4* stands out as an erroneous test case, and hence has no connections with other vertices.

