# OpenReview forum: "Enhancing Large Language Models in Coding Through Multi-Perspective Self-Consistency"
_ICLR.cc/2024/Conference — Submitted to ICLR 2024_

### Official Review · Reviewer_F87r · 2023-10-28

**Soundness:** 3 good
**Presentation:** 2 fair
**Contribution:** 2 fair
**Rating:** 5
**Confidence:** 4

**Summary:**

This paper proposes to improve language models' code generation via multi-perspective self-consistency.

Specifically, the multi-perspective is defined as sampling generations from 1) solutions; 2) specifications; 3) test cases. Then the authors propose to construct a graph based on those samples, and use inter-consistency and intra-consistency (via solving an optimization problem) to improve the final generations.

The authors did experiments on a few language models and show their method outperform several baselines.

**Strengths:**

- The authors did a thorough exploration on what factors could potentially affect code generation performances in LLMs, and it's interesting to see that all three perspectives, solution, specification, and test cases all play important roles in improving the final performance.

- The experiments are done on a fairly comprehensive set of code generation models, including GPT, Code Llama and WizardCoder.

- For intra-consistency, the authors also did a comprehensive study over what kind of measure functions can help the most.

**Weaknesses:**

- Novelty: [1] already proposed to construct a graph and explore more fine-grained consistency via MAX-SAT solving (although for natural language based reasoning tasks). The similarities and differences should be better discussed in the current paper.

[1] Jung et al. Maieutic prompting: Logically consistent reasoning with recursive explanations. EMNLP 2022.

- Generalizability: although framed as "multi-perspectives", this current paper only explores a single use-case of code generation, and with very specific perspectives: solution, specification, and test cases. It would be interesting to show if this method can be generalized to other tasks (e.g., math tasks, commonsense reasoning, or symbolic reasoning).

- Added complexity and ad-hoc design choices: The current framing adds a lot of complexity to the existing baselines, and many of the design choices are not well justified. In practice it would be difficult to deploy such a method to ensure optimal performance. E.g.,
1) designing the perspectives: for each task, how much manual effort is needed to design those perspectives? (and writing the prompts for each perspective?) How sensitive is the final performance w.r.t. the prompts written for each perspective?
2) constructing the graph: each edge needs to be specifically designed (section 3.2), why those choices and how much difference does it make if designed differently? For intra-consistency, similarly the authors designed many different measures, and based on the experiment results the best measure varies depending on the task. How would one pick which measure to use in practice?
3) solving the optimization: depending on the number of nodes and edges, the solving part could be very expensive; even with the iterative algorithm, it might take many rounds to reach convergence. This point needs to be better discussed.
4) choice of parameters: what is the value of \alpha in experiments? From Figure 4, on human-eval the performance varies a lot depending on \alpha. The final reported performance seems to be the \alpha that achieves the best performance. But without a training/dev set, how could one pick \alpha in an unsupervised way?

- Fair evaluation: Table 2 shows the proposed method yields some gains over several baselines. But digging deeper, a fair comparison should be made between methods that use *the same number of generated samples*. MPSC uses many more samples (200 solutions, 100 specifications, 500 test cases) while most of the baselines only use solutions (200 samples only). In addition, MPSC-label is not a fair comparison given it uses human labels.
1) if given the same number of samples, i.e., baselines with 800 samples vs MPSC, how does the performance compare?
2) what is the variance of the proposed method, given that more samples from different perspectives are drawn?
3) how does the proposed method compare with [2]? [2] also uses test cases to improve language models' code generation.
4) Table 4 shows the test cases give the most gains, so maybe a simple baseline could be added: use the generated test cases to filter out incorrect generated solutions, and then apply self-consistency on the filtered set.

[2] Chen et al. Teaching Large Language Models to Self-Debug. 2023.

**Questions:**

- How is the value of \alpha chosen for each task? Is it chosen based on the final performance (which means the test data is known when you pick \alpha)?

- How does the proposed method work on the best model experimented (GPT-4)? Does it still give gains?

---

> ### Author Response · Authors · 2023-11-18
> **Response to Reviewer F87r (1/2)**
>
> We really appreciate your feedback and suggestions, but we have some different opinions on the comments. We provide explanations to address your concerns and questions. We sincerely hope you can reconsider the review after reading our responses.
>
> > [1] already proposed to construct a graph and explore more fine-grained consistency via MAX-SAT solving
>
> Thanks for your suggestions about the missing related work.
>
> Acknowledging that both MPSC and Maieutic Prompting are post-hoc methods aimed at enhancing the reasoning abilities of LLM, it is crucial to highlight the distinctions between them.
>
> The primary difference lies in motivation. Maieutic Prompting addresses the inconsistency between the explanation and label generated by LLM, focusing on **the two-value entailment relation (T/F) between a statement and an explanation**. In contrast, MPSC is designed to fully exploit **the consistency among  outputs from different perspectives** within LLM. In our opinion, MPSC covers a broader scope, as it is straightforward to regard the statement and explanation as two perspectives and leverage the entailment relation as a form of inter-consistency measure within our framework.
>
> Furthermore, there are many differences in the specific method design. Maieutic Prompting regards the generated outputs as a tree structure, emphasizing the entailment between each parent and son. Therefore, its outputs has a sequential causal relationship. In contrast, in our approach, outputs from different perspectives has no order. And we construct a graph to encode pair-wise relations between vertices from different perspectives. Regarding the inference process,  while Maieutic Prompting treats it as a MAX-SAT problem, which is NP-hard, we propose a continuous optimization problem over the graph, benefiting from a closed-form solution.
>
> Thanks for pointing out this paper. We will add it into the related work session in the updated paper.
>
> > Generalizability...
>
> We have made a general response about your concerns. Please refer to it for more details.
>
> > how much manual effort is needed to design those perspectives? (and writing the prompts for each perspective?) How sensitive is the final performance w.r.t. the prompts written for each perspective?
>
> In response to the first question about manual designed perspectives, we have made a general response. Please refer to it for more details.
>
> Regarding the other two questions about prompt design, we barely work on prompt design and only use convention natural language instructions (please refer to Table 12/13/14). It is evident that the prompt we used is definitely not (even far from) the optimal.
> Moreover, we emphasize that the prompt design will not only affect MPSC but also other baselines, since they utilize the same set of generated solutions and test cases.
>
> In our opinion, the two points can guarantee the reproducibility of the reported performance in our paper and the prompt insensitivity of MPSC.
>
> > constructing the graph: each edge needs to be specifically designed (section 3.2), why those choices and how much difference does it make if designed differently?
>
> The Solution, Specification and Test case are three well-established perspectives for code generation [1]. Based on their definitions, the inter-consistency measure through code execution is pretty natural and intuitive. We also discuss the possibility of the automatic inter-consistency measure in the general response. Please refer to it for more details.
>
> [1] Agile Software Development Methods: Review and Analysis
>
> > For intra-consistency, similarly the authors designed many different measures, and based on the experiment results the best measure varies depending on the task. How would one pick which measure to use in practice?
>
> We designed many different measures to better analyze different types of intra-consistency within LLM (e.g. *Bayes Risk* is related to lexical consistency, *Cardinality* is designed to evaluate semantic consistency). Across various scenarios, we consistently observed robust performance with MPSC-Weighted Cardinality, making it our recommended preference.
>
> > solving the optimization: depending on the number of nodes and edges, the solving part could be very expensive
>
> The iterative algorithm converges pretty fast. On HumanEval, the average convergence is achieved in 45.21 iterations, while on mbpp, it takes 32.52 iterations. Notably, the algorithm converges even more swiftly on CodeContests, requiring only 11.75 iterations. This consistent pattern of fast convergence across various benchmarks underscores the algorithm's efficiency and effectiveness.

---

> ### Author Response · Authors · 2023-11-18
> **Response to Reviewer F87r (2/2)**
>
> > without a training/dev set, how could one pick \alpha in an unsupervised way?
>
> First of all, we want to emphasize that even with randomly selected alpha (e.g. 0.05 ~ 0.95), the performance of MPSC (especially the Weighted Cardinality setting) still outperforms baselines (please refer to Figure 4).
>
> We admit that we cannot select the optimal alpha without priori about the test data distribution. In our main experiment, we sample 10 cases (which is a relatively small number) as our dev dataset. We then conduct a grid search for alpha on it (from 0.05 to 0.99 with a step of 0.05).
>
> > if given the same number of samples, i.e., baselines with 800 samples vs MPSC, how does the performance compare?
>
> We totally agree with you about this fair experimental settings. In response to your concerns, we reduce the number of samples we use to 80 solutions, 20 specifications and 100 test cases, which is even unfair for MPSC since we can actually extract 10 test cases per API call. Because of your attention to variance in our method, we conducted sampling with five distinct seeds and reported the mean Pass@1 performance along with the std. The results are shown as follows. It is evident that MPSC still maintains high performance.
> - 83.52(±0.72) on HumanEval
> - 73.43(±0.43) on HumanEval+
> - 8.89(±0.7) on CodeContests
> - 72.2(±0.76) on MBPP
>
> > what is the variance of the proposed method, given that more samples from different perspectives are drawn?
>
> We think the variance is likely to decrease, since the only source of randomness in MPSC comes from the sampling process of LLM. As the number of samples increases, it is reasonable to anticipate a reduction in variation.
>
> > how does the proposed method compare with “Teaching Large Language Models to Self-Debug.”
>
> This paper doesn’t release their codes, which makes the reproduction much harder. We have already tried to reproduce the method according to the prompts given in their appendix, but the gains is pretty small on our datasets.
>
> > Table 4 shows the test cases give the most gains, so maybe a simple baseline could be added: use the generated test cases to filter out incorrect generated solutions, and then apply self-consistency on the filtered set.
>
> Even though the test case perspective gives the most gains in ablation study, it doesn’t guarantee the quality of generated test cases. Notably, the accuracy of generated test cases is relatively low, with only 63.83% on HumanEval and a mere 24.54% on MBPP. Directly filtering solutions through these generated test cases is anticipated to result in subpar performance, potentially even worse than employing vanilla self-consistency without the filtering.
>
> We want to reiterate that the substantial performance gains of MPSC are not attributed to the quality of the LLM generated outputs. Instead, these gains stem from the consistency within the LLM, which is fully exploited by MPSC.
>
> > How does the proposed method work on the best model experimented (GPT-4)? Does it still give gains?
>
> We are sorry that GPT4 API is too expensive to conduct the experiments as you wish. Here we conduct an alternative experiment to improve GPT4 generated solutions with ChatGPT generated test cases and specifications. The Pass@1 results are shown below. We use the same alpha as in the main experiment.
>
> | Method|HumanEval|HumanEval+|CodeContests|MBPP|
> | ------------------------- | ----- | ----- | ----- | ----- |
> | GPT4                      | 81.55 | 71.43 | 6.07  | 71.26 |
> | MPSC-Uniform              | 89.02 | 75.89 | 7.88  | 74.79 |
> | MPSC-MBR                  | 89.02 | 78.66 | 8.48  | 73.24 |
> | MPSC-Weighted Cardinality | 89.63 | 78.01 | 10.39 | 74.67 |
>
> The efficacy of MPSC remains evident in its substantial performance gains, even when applied to GPT-4. It should be noted that these gains, while considerable, appear relatively modest compared with the remarkable improvements seen in ChatGPT. This discrepancy is understandable given that MPSC relies on the consistency within a single LLM. Despite ChatGPT and GPT-4 are two versions of the same LLM, subtle variations can still influence outcomes. We posit that employing MPSC with outputs exclusively generated by GPT-4 could potentially unlock unparalleled performance.

---

> > ### Comment · Reviewer_F87r · 2023-11-22
> >
> > Thanks for the response. Overall I think the proposed method still requires too much manual design: if only one or two of them is manually designed like prompting, it might be fine, but MPSC requires many other manual choices as well, and the final performance varies quite a lot depending on each choice, i.e., there's no single universal best choice people can use in practice. Also picking \alpha requires a dev set which is an unfair comparison to existing methods that are entirely unsupervised. Hence I will keep my original rating.

---

> ### Author Response · Authors · 2023-11-23
> **Response to Reviewer F87r**
>
> Thanks again for your comments and willingness for discussion.
>
> We acknowledge your opinion that we do include manual design into MPSC for code generation, including the perspective and inter-consistency measure design. But as stated in the general response, we want to again underline that MPSC is inherently capable of entirely automatic reasoning without manual efforts. And in our opinion, **leveraging human priori about the task (i.e. the well-defined perspectives and verification in software engineering) for better reasoning of LLM should not be regarded as a drawback of our work**. With the current progress of LLM, achieving significant performance gains (e.g. surpassing GPT4 with GPT3.5) while requiring no manual efforts (not even defining one singular consistency function) is not feasible. Also, prompt engineering can serve as a good example of unleashing great potentials of LLM with a few manual efforts, which is valuable for real application.
>
> Regarding the choice of alpha, we only acknowledge that we cannot choose the optimal alpha without any information about data distribution (and only 10 samples can give a good estimation). We must emphasize that, even with the worst alpha, MPSC-Weighted Cardinality still surpasses other baselines as shown in Figure 4. And we recommend to use MPSC-Weighted Cardinality as stated in our previous comments.

---

### Official Review · Reviewer_cqa4 · 2023-10-31

**Soundness:** 3 good
**Presentation:** 2 fair
**Contribution:** 2 fair
**Rating:** 5
**Confidence:** 4

**Summary:**

This paper investigates the intra-consistency and inter-consistency of LLMs through the code generation tasks. A multi-perspective self-consistency (MPSC) framework is proposed to enhance the decoding process of LLMs by introducing a multipartite graph. MPSC achieve noticeable improvement in Pass@1 on four code generation tasks.

**Strengths:**

1. This work investigates both intra-consistency and inter-consistency of LLMs
2. The proposed MPSC achieves significant improvement on code generation
3. The proposed MPSC can act like a plug-in to enhance other LLMs

**Weaknesses:**

1. The MPSC is complicate and unstable which limits the reproducibility
2. The cost of MPSC is very high which limits its application
3. The narrative of the paper is not clear

**Questions:**

Comments
1. There have been some works investigate the inter-consistency issue [1]. I think the inter-consistency issue should come from different LLMs other than a single LLM. Therefore, the inconsistency among solutions, specifications, and test cases seems like a kind of intra-consistency within a single LLM.

[1] Examining Inter-Consistency of Large Language Models Collaboration: An In-depth Analysis via Debate

2. MPSC needs to sample 200 solutions for a problem, a very high temperature should be used to ensure diversity. I am a little worried about its reproducibility and stability. Moreover, it is costly to implement MPSC due to we should sample 200 solutions for each problem.

3. I have spent a lot of time trying to understand the details of the MPSC framework. A real example of the multipartite graph will be helpful to lower the threshold of understanding.

---

> ### Author Response · Authors · 2023-11-17
> **Response to Reviewer cqa4**
>
> We really appreciate your feedback and suggestion, but we have some different opinions on the comments. We provide explanations to address your concerns and questions. We sincerely hope you can reconsider the review after reading our responses.
>
> > 1. The MPSC is complicate and unstable which limits the reproducibility
>
> We are sorry if our description of MPSC is hard to understand, which may cause your confusion. But we want to emphasize that **MPSC is a simple, intuitive and deterministic algorithm.**
>
> Firstly, MPSC requires the LLM to generate outputs from different perspectives. And intuitively we can leverage graph to model the relations among them, in which the edge weight w(vi,vj) is well defined as the inter-consistency between two vertices.
>
> With the graph encoding the consistency information, we can then easily employ a simple but effective graph ranking algorithm (i.e. manifold ranking [1]) to aggregate the information and hence select the most consistent solution.
>
> Both the graph construction process (edge weight is obtained through code execution) and the graph ranking process are deterministic. So the reproducibility can be guaranteed.
>
> [1] Learning with Local and Global Consistency
>
> > 2. There have been some works investigate the inter-consistency issue [2]. I think the inter-consistency issue should come from different LLMs other than a single LLM.
>
> Thanks for your thoughtful advice about the missing related work.
>
> We need to emphasize that the inter-consistency defined in this work is among different perspectives with respect to one query / question.
>
> We believe that, for a single LLM, there exists a form of consistency in its responses to the same question across different perspectives, which we term inter-consistency. Simultaneously, there is intra-consistency observable in various responses emerging from the same perspective. MPSC is designed to fully exploit both these two kinds of consistency information within one model.
>
> In essence, our emphasis lies in leveraging the inter-consistency of the same LLM. On the contrary, [2] finds the inter-**IN**consistency between different models, and proposes to solve it through a formal debate framework. Even though we happen to use the same name of “inter-consistency”, we refer to different concepts. And their motivation and contribution are quite different from ours.
>
> Thanks for pointing out this paper. We will consider adding it into the related work session in the revision version.
>
> [2] Examining the Inter-Consistency of Large Language Models: An In-depth Analysis via Debate
>
> > 3. The cost of MPSC is very high which limits its application.
>  MPSC needs to sample 200 solutions for a problem, a very high temperature should be used to ensure diversity. I am a little worried about its reproducibility and stability. Moreover, it is costly to implement MPSC due to we should sample 200 solutions for each problem.
>
> First, we want to clarify that sampling 200 solutions for evaluation has been a convention setting in code generation since Codex [3]. And we follow the same temperature setting as in CodeT [4].
>
> To fully address your concern about reproducibility, we resample solutions with 3 random seeds and report the mean Pass@1 performance with std. We only experiment with WizardCoder(7b/13b/34b) on HumanEval here, because the OpenAI API access is both expensive and time-consuming during this period. The results are shown below. It is evident that the improvement $\Delta$ brought by MPSC is quite stable.
>
> |Model Size| Random Selection | MPSC |$\Delta$|
> |--|--|--|--|
> |7B|52.20(±2.16)|70.97(±0.93)|18.77(±1.74)|
> |13B|58.62(±1.22)|76.33(±0.94)|17.71(±2.17)|
> |34B|66.04(±1.27)|79.99(±0.72)|13.95(±1.83)|
>
> We are also grateful for your careful consideration about the cost of MPSC in practical scenarios. Therefore, we reduce the solution sampling number into 10 / 50 / 100. The results are shown below. Even with fewer solutions, MPSC can still accurately select the correct ones (if exists) and achieve extraordinary performance.
>
> |#Samples|Method|HumanEval|HumanEval+|MBPP|CodeContests|
> |--|--|:--|:--|:--|:--|
> |100|random selection|68.38(±0.08)|58.70(±0.12)|66.82(±0.07)|2.51(±0.07)|
> |100|MPSC|85.61(±0.3)|76.08(±0.33)|72.19(±0.17)|10.87(±0.29)|
> |50|random selection|68.50(±0.24)|58.78(±0.27)|66.85(±0.09)|2.56(±0.06)|
> |50|MPSC|85.13(±0.3)|75.75(±0.49)|72.14(±0.36)|10.68(±0.46)|
> |10|random selection|68.40(±0.6)|58.85(±0.49)|67.02(±0.12)|2.81(±0.21)|
> |10|MPSC|83.42(±0.9)|73.79(±0.53)|72.39(±0.24)|8.73(±1.98)|
>
> [3] Evaluating Large Language Models Trained on Code
>
> [4] CodeT: Code Generation with Generated Tests
>
> >  4. A real example of the multipartite graph will be helpful to lower the threshold of understanding.
>
> Again, we are sorry for the unclear description of MPSC. Thanks for your advice, we will add a real example of the graph in the revision version.

---

> > ### Comment · Reviewer_cqa4 · 2023-11-22
> >
> > Thanks for the authors' response. However, I will keep my score and believe that the paper can be further improved and clarified.

---

> ### Author Response · Authors · 2023-11-22
> **Thanks for Comments and Willingness for Further Discussion**
>
> Thank you for sharing your comments, and we appreciate your perspective. Do you have any other questions or additional concerns? we are willing to have further discussion. Your thoughtful advice is valuable to our refinement, and we welcome any additional insights you may have.

---

### Official Review · Reviewer_F8qr · 2023-11-01

**Soundness:** 1 poor
**Presentation:** 3 good
**Contribution:** 2 fair
**Rating:** 5
**Confidence:** 5

**Summary:**

This paper introduces a framework based on multiple perspectives for further researching and enhancing the previous self-consistency method. The idea is to divide self-consistency into consistency among various perspectives. The authors have validated the effectiveness of this method on different models and programming datasets.

**Strengths:**

1. The improvement over the baselines is significant. However, it's unclear whether this improvement is due to the allowance of code execution.

2. The paper is easy to read.

3. The method proposed in this paper can be applied to both closed-source large models (like ChatGPT) and open-source large models.

**Weaknesses:**

1. A key issue with the methodology and experiments of this paper is that the proposed method actually requires the execution of code, rather than merely generating it. For instance, in Listing 3 of Appendix 3, one needs to execute the solution to complete the validation with test cases. The authors need to emphasize this distinction from other methods. Clearly, when code execution is allowed, the task becomes easier. This issue makes the main experimental results of the paper, as shown in Table 2, unfair, as other methods do not require code execution.

2. Can the method proposed in this paper be applied under conditions where code execution is not allowed? If so, what are the results? The authors have not demonstrated this.

3. Is the learning approach designed in this paper overly complicated? Is it possible to avoid using w(vi,vj) and f(vi) and directly employ an MLP or other ensemble methods to obtain the answer? For instance, self-consistency actually uses max-vote directly. Overly complex optimization algorithms make the methodological contributions of this paper ambiguous.

4. The specific "perspectives" used in this paper do not entirely align with intuition. For example, in Figure 2, the paper views the solution, specification, and test case as three perspectives, whereas conceptually, they are three answer components.

**Questions:**

Please refer to the weaknesses.

---

> ### Author Response · Authors · 2023-11-17
> **Response to Reviewer F8qr**
>
> We really appreciate your feedback, but we have some different opinions on the comments. We provide more explanations for each weakness you pointed out in your comments. We sincerely hope you can reconsider the review after reading our responses.
>
> > 1. A key issue with the methodology is that the proposed method actually requires the execution of code ... Clearly, when code execution is allowed, the task becomes easier. This issue makes the main experimental results ... unfair, as other methods do not require code execution.
> 2. Can the method proposed in this paper be applied under conditions where code execution is not allowed?
>
> On one hand, we never execute codes with golden test cases, but only with generated test cases. It doesn’t make the task easier because the correctness of generated test cases is not guaranteed. On the other hand, main baselines (including Self-Consistency, MBR-Exec, CodeT) **all require code execution** . Therefore, the comparison in our main experiment is fair.
>
> - Self-Consistency: we follow the setting in [1]. If two solution pass the same set of generated test cases and specifications, we regard them “consistent”. Then we take a majority voting to rank solutions following [2].
> - CodeT: CodeT first uses generated test cases to verify each solution by code execution. Then it utilizes RANSAC algorithm to create consensus sets based on execution results. The size of consensus set is then used to rank solutions.
> - MBR-Exec: This method ranks solutions by minimum Bayes risk decoding based on the execution results in the generated test cases.
>
> We will make the details of these baselines all clear in our updated paper.
>
> [1] CodeT: Code Generation with Generated Tests
> [2] Self-Consistency Improves Chain of Thought Reasoning in Language Models
>
> > 3. Is the learning approach designed in this paper overly complicated? Is it possible to avoid using w(vi,vj) and f(vi) and directly employ an MLP or other ensemble methods to obtain the answer?
>
> We want to argue that the design of MPSC is simple and intuitive.
>
> f(vi) is the ranking score of each output, which is necessary for any ranking algorithm since it is the final objective. The edge weight w(vi,vj) is naturally defined as the inter-consistency of two vertices from different perspectives in the graph, which can be deterministically obtained by code execution. After we build the graph (we will add a real example of the multipartite graph in our paper for clearer understanding), we can easily apply a simple but effective graph ranking algorithm here, i.e. manifold ranking [3], to aggregate the consistency information and select the best solution.
>
> It is clear that **MPSC is not a supervised learning approach**. It’s a non-parametric, deterministic algorithm. We don’t need to involve any learnable parameters, e.g. using MLP.
>
> There indeed exists other ensemble strategies like self-consistency, and we have included it as baseline. MPSC surpasses it by a large margin.
>
> [3] Learning with Local and Global Consistency
>
> > 4. The specific "perspectives" used in this paper do not entirely align with intuition. For example, in Figure 2, the paper views the solution, specification, and test case as three perspectives, whereas conceptually, they are three answer components.
>
> Solution, test case and specification are three important aspects in agile software development methods [4]. They play different roles in the software development lifecycle. They are not different “answer components” because the only answer component required in code generation task is the solution. One can certainly complete this task without introducing test cases and specifications. And our key insight is to make LLM think from these non-essential but significant perspectives, so that we can leverage the consistency information within LLM to further enhance its own capabilities.
>
> [4] Agile Software Development Methods: Review and Analysis

---

> ### Comment · Reviewer_F8qr · 2023-11-22
>
> Thank you for your response. Now I understand that other baselines also employ code execution. This should be emphasized, otherwise readers might mistakenly think that, like self-consistency, it only requires Large Language Models (LLMs). However, I noticed that the main text of the paper did not mention this.
>
> Considering that MPSC requires an environment for code execution, this raises some new concerns for me:
>
> 1. Under this limitation, the contribution of MPSC seems diminished. For instance, given that CodeT has already used LLMs to generate test cases, the additional generation of Solution and Specification in this paper does not appear to be a significant innovation.
>
> 2. The original paper on Self-Consistency was not applied to code, nor did it require code execution. Therefore, adapting Self-Consistency to code might not be straightforward. This makes the comparison between MPSC and Self-Consistency somewhat unclear in terms of its meaning.
>
> In summary, although the authors have addressed some of my concerns, they have also raised new ones, especially regarding the novelty compared to existing methods like CodeT. Additionally, the methodology section (Sec 3) of MPSC still seems overly complex.

---

> ### Author Response · Authors · 2023-11-23
> **Response to Reviewer F8qr**
>
> Thanks for your comments. We have already added the implementation details of baselines in Appendix E.
>
> Regarding the contributions and novelty of MPSC, we have different opinions. The main contribution of MPSC's is fully leveraging consistency within LLM through an optimization over graph, not using code execution on test cases. The same thing can also apply to CodeT. Its contribution comes from using RANSAC algorithm in code generation. Its novelty doesn't diminish because previous work has already used LLM to generate test cases (Alphacode[1]) and introduced code execution (Alphacode, MBR-exec[2]).
>
> Self-consistency cannot be directly used for code generation (any NLG tasks in fact), as stated in their paper "One should note that self-consistency can be applied only to problems where the ﬁnal answer is from a ﬁxed answer set". Our implementation of Self-consistency makes it available for code generation and fair for comparison.
>
> [1] Competition-level code generation with AlphaCode
> [2] Natural Language to Code Translation with Execution

---

### Official Review · Reviewer_myx4 · 2023-11-06

**Soundness:** 3 good
**Presentation:** 3 good
**Contribution:** 3 good
**Rating:** 6
**Confidence:** 5

**Summary:**

This paper proposes Multi-Perspective Self-Consistency (MPSC), which extends the original self-consistency framework to consider both inter-consistency and intra-consistency. For code generation, they consider 3 perspectives: solutions, specifications and test cases. MPSC generates multiple samples for each perspective, then constructs a multipartite graph and learns a scoring function to select the final answers. They compare MPSC to other baselines including self-consistency, MBR-Exec, CodeT and self-collaboration. They demonstrate that MPSC consistently outperforms the baselines by a significant margin on multiple coding benchmarks.

**Strengths:**

1. MPSC is a natural extension of self-consistency for code generation, where the consistency among the solution, test cases and specifications can be precisely verified by code execution.

2. The experiments show remarkable performance improvement compared to strong baselines that utilize multiple samples and/or test cases.

**Weaknesses:**

The main weaknesses of this work are: (1) the implementation details are unclear; and (2) some ablation studies are missing. Specifically, I have the following questions:

1. How is MBR-Exec implemented? I do not understand why MBR-Exec can perform worse than self-consistency. To my understanding, self-consistency selects the final program based on the exact match; i.e., selecting the most frequently appeared code in all samples. On the other hand, MBR-Exec selects programs based on the frequency of execution results. Does MBR-Exec utilize the given test cases as in the original paper?

2. For MPSC-Label, how are the golden test cases utilized? Do you directly filter out those programs that do not pass the test cases? In general I do not understand why MPSC-Weighted Cardinality can sometimes outperform MPSC-Label.

3. It is interesting to see that GPT-3.5 with MPSC can outperform GPT-4. However, sampling 200 solutions is still very expensive. Do you have results with fewer number of samples, e.g., 10 or 100? What is pass@200, which should be the upper bound of the performance?

4. It is helpful to add discussion on the quality of generated test cases and specifications. For example, what are the true positive and false negative rates?

Also, I think MPSC is well-suited for code generation, but how to extend it to other domains remains unclear.

**Questions:**

1. How is MBR-Exec implemented? Does MBR-Exec utilize the given test cases as in the original paper?

2. For MPSC-Label, how are the golden test cases utilized? Do you directly filter out those programs that do not pass the test cases?

3. Do you have results with fewer number of solutions, e.g., 10 or 100 instead of 200? What is pass@200, which should be the upper bound of the performance?

4. It is helpful to add discussion on the quality of generated test cases and specifications. For example, what are the true positive and false negative rates?

---

> ### Author Response · Authors · 2023-11-17
> **Response to Reviewer myx4**
>
> We are grateful for your positive feedback and thoughtful advice. We address the weaknesses and questions you pointed out as follows. We hope this can resolve your doubts.
> >1. How is MBR-Exec implemented? Does MBR-Exec utilize the given test cases as in the original paper?
>
> We are sorry for the missing implementation details of baselines.
>
> For the self-consistency baseline, we actually don’t use the lexical exact match, but follow the setting in [1]. That is, if two solutions pass the same set of generated test cases and specifications, we regard them “consistent”. Then we take a majority voting to rank solutions.
>
> For MBR-exec implementation, we don’t use the golden test cases but the generated test cases for a fair comparison. Because MPSC and other baselines do not execute the golden test cases.
>
> So it is reasonable that self-consistency exceeds MBR-Exec. We will make these details clearer in the updated paper.
>
> [1] CodeT: Code Generation with Generated Tests
>
> >2. For MPSC-Label, how are the golden test cases utilized? Do you directly filter out those programs that do not pass the test cases?
>
> We are sorry for the misuse of phrase “golden test case”, which may cause your confusion.
>
> To avoid misunderstanding, we want to emphasize that labels used in MPSC-Label setting is the example test cases provided in docstrings (silver test cases), not the test cases used during testing (golden test cases).
>
> We **NEVER** directly filter the programs with silver test cases or golden test cases.
>
> The main idea of MPSC-Label (we refer you to the section 3.3 and Table 1 for more details) is to pass the messages of silver test case vertices to the generated solution vertices through the graph. On the other hand, MPSC-Weighted Cardinality utilizes the intra-consistency within LLM.
>
> They utilize different kinds of information (one from the outsides, one from the insides). The performance is surprising but also reasonable. As shown in [2], the intra-consistency is highly correlated to correctness. Our experiment further demonstrates this point.
>
> [2] Self-Consistency Improves Chain of Thought Reasoning in Language Models
>
> >3. Do you have results with fewer number of solutions, e.g., 10 or 100 instead of 200? What is pass@200, which should be the upper bound of the performance?
>
> First, we want to clarify that sampling 200 solutions for evaluation has been a convention setting in code generation since Codex [3].
>
> We are grateful for your careful consideration about the costs of MPSC in practical scenarios. Here we add an extensive experiment by only sampling 10 / 50/ 100 solutions (we conduct sampling for 5 times with different seeds, and report mean Pass@1 performance with std error bar). We use the same MPSC-Weighted Cardinality settings as that in the main experiment. It is evident that even with fewer solutions samples, MPSC is able to select the correct ones (if exists), and outperforms baseline (i.e. random selection) by a significant margin.
>
>
> |#Samples|Method|HumanEval|HumanEval+|MBPP|CodeContests|
> |--|--|:--|:--|:--|:--|
> |100|random selection|68.38(±0.08)|58.70(±0.12)|66.82(±0.07)|2.51(±0.07)|
> |100|MPSC|85.61(±0.3)|76.08(±0.33)|72.19(±0.17)|10.87(±0.29)|
> |50|random selection|68.50(±0.24)|58.78(±0.27)|66.85(±0.09)|2.56(±0.06)|
> |50|MPSC|85.13(±0.3)|75.75(±0.49)|72.14(±0.36)|10.68(±0.46)|
> |10|random selection|68.40(±0.6)|58.85(±0.49)|67.02(±0.12)|2.81(±0.21)|
> |10|MPSC|83.42(±0.9)|73.79(±0.53)|72.39(±0.24)|8.73(±1.98)|
>
> Here we also report pass@200, which is the upper bound of all methods. The gap between the upper bound and MPSC is relatively small (8% on HumanEval, 5% on CodeContests, 10% on MBPP), and again proves the effectiveness of our method.
> - HumanEval: 93.90
> - HumanEval+: 90.24
> - CodeContests: 19.39
> - MBPP: 83.37
>
> [3] Evaluating Large Language Models Trained on Code
>
> >4. It is helpful to add discussion on the quality of generated test cases and specifications.
>
> We have already discussed the quality of generated test cases and specifications in Table 10. Instead of using accuracy, we utilize pass@k to demonstrate the effectiveness of MPSC on ranking other perspectives (i.e. test cases and specifications). And the pass@1 is actually equal to accuracy. So the accuracy of specifications and testcases are as following:
>
> - HumanEval: Test case 63.83%, Specification 45.66%
> - MBPP: Test case 24.54%, Specification 50.83%
>
> It should be noted that the accuracy of specification is not 100% accurate. Because the formal verification of specification requires professional human evaluation, which is both expensive and time-consuming. As an alternative, we apply a looser verification by examining whether a specification can pass golden test cases.
>
> >5. how to extend it to other domains remains unclear.
>
> Please refer to the general response for more details.

---

> > ### Comment · Reviewer_myx4 · 2023-11-23
> >
> > Thank the authors for the explanation, it clarifies the settings of some experiments. I keep my review score, though I think extra work should still be done to make the approach design and implementation details clearer in paper writing.

---

### Official Review · Reviewer_q6tm · 2023-11-08

**Soundness:** 3 good
**Presentation:** 3 good
**Contribution:** 2 fair
**Rating:** 5
**Confidence:** 4

**Summary:**

This paper presents Multi-Perspective Self-Consistency (MPSC), a novel framework aiming at improving the performance of LLMs at decoding time in complex reasoning tasks such as code generation. Extending from previous work on self consistency (e.g., Wang et al., 2022), the authors introduce multiple perspectives and a formulation of inter-consistency, which captures the agreement between generated outputs from diverse perspectives. The authors conduct experiments in code generation using various code competition benchmarks and three perspectives: solution, specification, and test cases. Empiricial results show a good amount of improvement over the baseline method.

**Strengths:**

- The multi-perspective method is well-motivated and well-suited for tasks like code generation.
- The authors conduct comprehensive evaluation and show significant performance improvement over various baselines.
- The paper is well-written.

**Weaknesses:**

- The main limitaiton of the work is the useability on a broader range of tasks. Despite MPSC is claimed to be task-agnostic, only code generation was presented in the paper, which greatly limits the impact of this work. On one hand, the authors only study code competition task, and it is unknown whether the framework can work in code generation in the wild. On the other hand, the authors should consider including at least one more NL task to demonstrate the extensibility of the framework.

- It is unclear whether and how well the framework can generalize towards more perspectives. In code generation, there are only three perspectives, which is quite limited. It would be great to think about and demonstrate that MPSC can work with arbitary number of perspectives.

- The perspectives are manually curated now, which can be a limitation for tasks with vague perspective definitions. It would be great to discuss whether manual curation of perspectives is required and if not how that would impact the end performance.

**Questions:**

- What would be the reason for degradation in MBPP?

- The improvement is diminishing when using a higher k in pass@k. Can the authors perform experiments with a much higher k (e.g. 100) and see the gap there?

- For Bayes Risk, why would using BLEU metrics preferred especially given the code generation task?

- Can the authors discuss the overhead added due to the graph introduced by MPSC?

---

> ### Author Response · Authors · 2023-11-18
> **Response to Reviewer q6tm (1/2)**
>
> We highly value your constructive feedback and wish to express our gratitude. But we would like to present our differing views on the issues you raised. We have provided detailed explanations in our responses for each question you pointed out, and we sincerely hope you may reconsider your review.
>
> > 1. The main limitaiton of the work is the useability on a broader range of tasks...
> 2. Whether manual curation of perspectives is required and if not how that would impact the end performance...
>
> We have written a general response to your concern. Please refer to the it about the details.
>
> > 3. It is unknown whether the framework can work in code generation in the wild.
>
> We have conduct comprehensive experiments on four popular benchmarks of code generation. MPSC has shown superior performance on all four of them and surpassed other methods (even GPT4) by a large margin. To our best knowledge, none of the current LLM-based code generation research, e.g., [1][2][3] conducts experiments of code generation in the wild. And there is also no high-quality such benchmarks. We would be glad to conduct experiments if there is any. In theory, MPSC can be applied for any scenarios in the wild provided that code execution is feasible.
>
> [1] Evaluating Large Language Models Trained on Code
> [2] WizardCoder: Empowering Code Large Language Models with Evol-Instruct
> [3] Code Llama: Open Foundation Models for Code
>
> > 4. It is unclear whether the framework can generalize towards more perspectives.
>
> To avoid potential misunderstanding, we want to underscore that the introduction of “perspectives” is to induce outputs reflecting LLM’s understanding of the query, and therefore better exploit the consistency within LLM. So the number of perspectives should be adapted to the requirements of the given task, rather than being rigidly constrained by an arbitrary number.
>
> Regarding your concern, we would like to discuss about the impact of the number of perspectives on MPSC. It should be noted that the number of perspectives only determines the partitions of the constructed graph. MPSC framework contains two stages: Graph Construction (GC) and Graph Ranking (GR). The GC stage requires to measure pair-wise inter-consistency between vertices from two different perspectives and intra-consistency of each vertex. They both are independent of the number of perspectives. The GR stage is essentially an optimization process over a graph (see Eq.3 for details). Notably, this optimization process, with a closed-form solution (see Appendix A for details), does not leverage the properties of a multi-partite graph and hence remains unaffected by the number of perspectives.
>
> > 5. The improvement is diminishing when using a higher k in pass@k. Can the authors perform experiments with a much higher k (e.g. 100)
>
> First we want to emphasize that pass@k with small k (e.g. 1,2,5) is much more important in practical scenario since users won’t expect to do a second selection from a total of 50 solutions. And current works about code generation models (including ChatGPT, GPT4, StarCoder, WizardCoder) only report pass@1 performance.
>
> The diminishing improvement is expected with higher k theoretically. An intuitive way to understand that is from the definition of pass@k, which is the probability of whether the selected $k$ samples out of 200 pass unit tests (Refer to Appendix C for more details). The lowerbound of selection, random selection (also our baseline), has a pass@k value of $1-\frac{\tbinom{200-c}{k}}{\tbinom{200}{k}}$ ($c$ is the number of correct solutions out of 200 solutions). While the upperbound of selection has a pass@k value of 100% (if there exist correct solutions). When k=100, c=6, even the lowerbound can achieve 99%. When k=50, c=10, the lowerbound can achieve 95%. The small gap between lower bound and upper bound makes the evaluation meaningless for all ranking algorithms.
>
> > 6. For Bayes Risk, why would using BLEU metrics preferred especially given the code generation task?
>
> We choose BLEU because the MPSC-Bayes Risk setting is designed to measure the **lexical** intra-consistency. And we utilize the BLEU metrics following the first MBR method in NLP [4].
>
> If you refer to the semantical consistency of codes, the structural equivalence (Sec 3.3) reflects some sorts of “MBR” with respect to the code execution results.
>
> [4] Minimum Bayes-Risk Decoding for Statistical Machine Translation

---

> ### Author Response · Authors · 2023-11-18
> **Response to Reviewer q6tm (2/2)**
>
> > 7.  What would be the reason for degradation in MBPP?
>
> We conduct a case study about the failure modes of MPSC where the foundation model do generate some correct solutions. We find some serious ambiguities in MBPP because of no example test cases provided in its docstrings. Here we list two typical kinds of ambiguity.
>
> - format ambiguity: The type of return value is not explicitly defined in the prompt. In the following case, LLM generate more than 50 “misguided” answers which return the correct solution but in a `tuple` format.
>     - prompt:
>
>         ```bash
>         def sort_sublists(input_list):
>         '''
>         Write a function to sort each sublist of strings in a given list of lists.
>         '''
>         ```
>
>     - MPSC selected solution:
>
>         ```bash
>         for sublist in input_list:
>             sublist.sort()
>         return input_list
>         ```
>
>     - golden test cases for testing:
>
>         ```bash
>         assert sort_sublists((["green", "orange"], ["black", "white"], ["white", "black", "orange"]))==[['green', 'orange'], ['black', 'white'], ['black', 'orange', 'white']]
>         ```
>
> - semantic ambiguity: The instruction in docstrings is not clear and contains ambiguity. In the following case, LLM generate more than 50 “misguided” answers which regard the `str` variable as a list because the description `given list of words` in docstrings. But it is in fact a string. It also generate more than 250 "misguided" testcases with `str` variable as a `list`.
>     - prompt
>
>         ```bash
>         def long_words(n, str):
>         '''
>         Write a function to find words that are longer than n characters from a given list of words.
>         '''
>         ```
>
>     - MPSC selected solution:
>
>         ```bash
>         words = []
>         for word in str:
>             if len(word) > n:
>                 words.append(word)
>         return words
>         ```
>
>     - golden solution:
>
>         ```bash
>         word_len = []
>         txt = str.split(" ")
>         for x in txt:
>             if len(x) > n:
>                 word_len.append(x)
>         return word_len
>         ```
>
>
> In both cases, LLM is seriously misguided to a different understanding of docstrings because of the ambiguity. MPSC is designed to fully exploit the consistency within LLM to select the most consistent solution, and hence is seriously influenced. In our opinion, the degradation is not only plausible, but also further justifies the design of our framework.
>
> > 8. Can the authors discuss the overhead added due to the graph introduced by MPSC?
>
> The overhead of MPSC mainly comes from the inter-consistency measurement through code execution. For verification between two vertices, the time is about 1e-4 sec (measured by executing one solution with one test case for 1000 times) in average. Following our main experimental setting (200 solutions & 500 test cases & 100 specifications), the sequential execution time is about $(200\times 500+200\times 100+500\times 100)\times 0.0001s=17s$. It’s worth noting that all the measurement can be executed in parallel. Moreover, as shown in Table 5&6, MPSC maintains the superior performance with only 10% test cases and specifications, which only requires 1.25s with sequential execution.

---

### Author Response · Authors · 2023-11-17
**General Response: Possibility of Generalization to Other Tasks and No Manual Efforts**

Dear Reviewers,

We appreciate all the insightful feedback regarding the generalization of our proposed framework MPSC to other tasks and the possibility of MPSC without any manual efforts. We would like to address your concerns and provide a more nuanced understanding of the challenges and considerations involved.

To avoid misunderstanding, we want to again review MPSC. It contains two stages: Graph Construction (GC) and Graph Ranking (GR). Only the GC stage involves task-specific semantics (including perspective definition and inter-consistency measures design), while the GR stage is intentionally designed to be task-agnostic, making no assumptions about task-specific semantics.

Regarding the GC stage, **there is a easy and general way to utilize MPSC to handle arbitrary reasoning tasks without any manual efforts**. That is, we require LLM to automatically determine which perspective to use, then require it to generate various outputs with respect to each perspective, and to score each pair of outputs as the inter-consistency measure. It is a straightforward method of MPSC framework and can serve as a lower bound performance.

However, this approach, relying solely on LLM, may bring uncertainty and instability. In our opinion, though exhibiting strong capabilities, current LLMs still lack the ability of controllable planning (i.e. determine which perspectives to use)[1][2]. Moreover, automatic inter-consistency measurement by LLMs highly depends on their understanding, which could be unstable. We consider improving these abilities of LLM as another research topic and leave them as future work.

In this paper, we introduce human defined perspectives and determined consistency criteria to mitigate the instability associated with fully automated approaches. This choice allows us to **focus on demonstrating the importance of leveraging consistency information from multiple perspectives**, which is the essential contribution of our framework.

That’s why we choose the code generation task as the initial application of MPSC. In software engineering area, Solution, Test case and Specification are three well-established perspectives for code generation [3]. And the process of measuring inter-consistency through code execution in this context is both natural and deterministic.

**The extraordinary performance on code generation has demonstrated the effectiveness of MPSC and the importance of multi-perspective consistency within LLM.** It’s our first step to enhance LLM in reasoning with MPSC. While we acknowledge that determined consistency criteria for certain reasoning tasks is not easy to find, relying solely on LLMs (i.e. the straightforward method mentioned above) is always feasible. The challenge lies in enhancing LLMs' planning abilities and semantic equivalence determination. We view this as an avenue for future research, and we commit to making our contributions and the generalization potential of MPSC more explicit in our updated paper.

[1] Large Language Models Still Can't Plan (A Benchmark for LLMs on Planning and Reasoning about Change)

[2] On the Planning Abilities of Large Language Models (A Critical Investigation with a Proposed Benchmark)

[3] Agile Software Development Methods: Review and Analysis

---

### Author Response · Authors · 2023-11-21
**Rebuttal Revision**

We have diligently addressed the constructive feedback from the reviewers in our revised submission for ICLR 2024. Key enhancements include:

- A detailed example of constructed graphs is added in Appendix F as suggested by reviewer cqa4.
- The implementation details of baselines is added in Appendix E.
- A discussion about the stability of MPSC is added in Appendix D.

We remain open to further refinement through more discussions with reviewers.

---

### Meta-Review · Area_Chair_Wg8f · 2023-12-11

**Metareview:**

This paper presents Multi-Perspective Self-Consistency (MPSC), a novel framework aiming at improving the performance of LLMs at decoding time in complex reasoning tasks such as code generation. Extending from previous work on intra-consistency, the authors introduce multiple perspectives (solution, specification, and test cases) and a formulation of inter-consistency, which captures the agreement between generated outputs from diverse perspectives. The authors conduct experiments in code generation showing a good amount of improvement over the baseline method.

Reviewers liked how the method is natural for code generation and its empirical strengths. On the other hand, they objected to its applicability beyond code generation, the added inference cost to generate multiple perspectives, the usability since more manual efforts were required to design the perspectives and novelty beyond existing work especially regarding CodeT, which already used two perspectives.

In the response, the authors made several good points such as the simplicity of the method, implementation details, and that the novelty is in how to use multiple perspectives. Most reviewers fully understood the main points of paper and provided their opinions.

**Justification For Why Not Higher Score:**

reviewer majority and no reviewer strongly favored accepting

**Justification For Why Not Lower Score:**

N/A

---

### Decision · Program_Chairs · 2024-01-16

Reject